# Anterograde signaling controls plastid transcription via sigma factors separately from nuclear photosynthesis genes

Youra Hwang [1,8], Soeun Han[1,5,8], Chan Yul Yoo [1,6,8], Liu Hong[1], Chenjiang You[1,7], Brandon H. Le[1], Hui Shi[2], Shangwei Zhong[3], Ute Hoecker[4], Xuemei Chen [1] & Meng Chen [1] ✉

Light initiates chloroplast biogenesis in *Arabidopsis* by eliminating PHYTOCHROME-INTERACTING transcription FACTORs (PIFs), which in turn de-represses nuclear photosynthesis genes, and synchronously, generates a nucleus-to-plastid (anterograde) signal that activates the plastid-encoded bacterial-type RNA polymerase (PEP) to transcribe plastid photosynthesis genes. However, the identity of the anterograde signal remains frustratingly elusive. The main challenge has been the difficulty to distinguish regulators from the plethora of necessary components for plastid transcription and other essential chloroplast functions, such as photosynthesis. Here, we show that the genome-wide induction of nuclear photosynthesis genes is insufficient to activate the PEP. PEP inhibition is imposed redundantly by multiple PIFs and requires PIF3's activator activity. Among the nuclear-encoded components of the PEP holoenzyme, we identify four light-inducible, PIF-repressed sigma factors as anterograde signals. Together, our results elucidate that light-dependent inhibition of PIFs activates plastid photosynthesis genes via sigma factors as anterograde signals in parallel with the induction of nuclear photosynthesis genes.

The control of organellar gene expression by the cell nucleus is critical to the cellular homeostasis of all eukaryotic organisms. In mammalian cells, the mitochondrial DNA encodes 13 proteins required for oxidative phosphorylation. The misregulation of mitochondrial genes results in or is associated with neurodegenerative diseases and cancer[1]. The plastids in plants encode substantially more genes; the plastome of the reference species *Arabidopsis thaliana* (*Arabidopsis*) harbors 85 protein-coding genes, including 46 genes essential for photosynthesis and therefore the plant's life[2,3]. Because over 90% of mitochondrial and plastid proteins are encoded by the nuclear genome[4], organellar gene expression relies on the expression of hundreds of nuclear-encoded gene products that participate in organellar transcription, post-transcriptional RNA processing, and translation. However, the mechanism of anterograde, i.e., nucleus-to-organelle, signaling by which the nucleus regulates organellar gene expression in response to developmental and environmental cues remains unclear. Anterograde signaling is often interpreted simply as the genetic dependency of organellar gene expression on the nuclear genome; but it is conceivable that not all nuclear-encoded organellar proteins are employed as signaling molecules. In fact, the main challenge to define

[1]Department of Botany and Plant Sciences, Institute for Integrative Genome Biology, University of California, Riverside 92521 CA, USA. [2]College of Life Sciences, Beijing Key Laboratory of Plant Gene Resources and Biotechnology for Carbon Reduction and Environmental Improvement, Capital Normal University, Beijing 100048, China. [3]State Key Laboratory of Protein and Plant Gene Research, School of Life Sciences, Peking University, Beijing 100871, China. [4]Institute for Plant Sciences and Cluster of Excellence on Plant Sciences (CEPLAS), Biocenter, University of Cologne, Cologne, Germany. [5]Present address: Section of Cell and Developmental Biology, University of California, San Diego, La Jolla 92093 CA, USA. [6]Present address: School of Biological Sciences, University of Utah, Salt Lake City 84112 UT, USA. [7]Present address: Institute of Plant Biology, School of Life Sciences, Fudan University, Shanghai 200438, China. [8]These authors contributed equally: Youra Hwang, Soeun Han, Chan Yul Yoo. ✉e-mail: meng.chen@ucr.edu

anterograde signaling has been the difficulty in teasing apart regulators from the battery of nuclear-encoded components required for organellar biogenesis and/or functions[5]. Compared with mitochondria, the regulation of gene expression in plastids is expected to be more complex. This is not only attributable to the greater number of genes encoded by the plastid genome, but more importantly, due to the ability of plastids to differentiate into various cell-/tissue-specific types with distinct morphologies and functions[6]. Plastid differentiation is closely associated with plastid gene expression and is ultimately controlled by the host cell's nucleus. As such, plastid differentiation presents a unique experimental paradigm to interrogate the mechanism of coordinating nuclear and organellar gene expression, particularly the nuclear control of organellar gene expression by anterograde signaling.

Chloroplasts are photosynthetically active plastids derived from undifferentiated proplastids in the meristematic cells. Chloroplast biogenesis is coupled with leaf development and the production of mesophyll cells – the cell type specialized for photosynthesis. Chloroplast biogenesis in angiosperms (flowering plants) also depends upon light[6]. In *Arabidopsis*, after seed germination under the ground or in darkness, seedlings adopt a dark-grown developmental program called skotomorphogenesis or etiolation, which promotes the elongation of the embryonic stem (hypocotyl) and inhibits the expansion of the embryonic leaves (cotyledons)[7]. Plastids in the cotyledon cells of dark-grown seedlings differentiate into nonphotosynthetic chloroplast precursors called etioplasts. Etioplasts are not green because chlorophyll biogenesis is blocked in the dark, as the conversion of protochlorophyllide to chlorophyllide a by protochlorophyllide oxidoreductase (POR) requires light[8]. During skotomorphogenesis, protochlorophyllide accumulates to high levels in etioplasts and, together with its associated POR, forms characteristic crystalline structures called prolamellar bodies. Upon exposure to light, seedlings switch to a light-grown developmental program called photomorphogenesis, which restricts hypocotyl elongation and instead promotes cotyledon expansion and leaf development[7]. The developmental switch from skotomorphogenesis to photomorphogenesis, called de-etiolation, is accompanied by the differentiation of nonphotosynthetic etioplasts into photosynthetic chloroplasts, thereby enabling seedlings to transition to autotrophic growth powered by photosynthesis. Making chloroplasts involves reorganizing the inner membrane system from prolamellar bodies to thylakoid membranes and building up the photosynthetic machinery, including the chlorophyll-containing light-harvesting complex for light perception, protein complexes for photosynthetic electron transport and ATP production – such as photosystems I and II, the cytochrome $b_6f$ complex, the NADH dehydrogenase-like complex, and ATP synthase – and Calvin cycle enzymes for carbon fixation. These photosynthetic components are encoded by both nuclear and plastid genes, referred to as photosynthesis-associated nuclear genes (PhANGs) and plastid genes (PhAPGs), respectively. The coordinated activation of PhANGs and PhAPGs is crucial for chloroplast biogenesis.

Chloroplast biogenesis is ultimately controlled by the host cell's nucleus[3]. The transition from etioplasts to chloroplasts during de-etiolation is initiated by light through photoreceptors, including the red/far-red light-sensing phytochromes (PHYs) and the blue-light-absorbing cryptochromes (CRYs)[7]. Photoactivated PHYs and CRYs are localized in the nucleus but not in plastids[9,10]. The primary action of light signaling in initiating photomorphogenesis is to reprogram the nuclear genome, including activating the nuclear photosynthesis program, or PhANGs. PHYs and CRYs control nuclear gene expression by regulating the stability and activity of a family of basic helix-loop-helix transcription factors named PHYTOCHROME-INTERACTING FACTORs (PIFs)[11–14]. In *Arabidopsis*, PIFs include eight members (PIF1–8), of which PIF1, PIF3, PIF4, and PIF5 collectively and redundantly repress PhANGs in the nucleus[15–17]. A dark-grown *pif1pif3pif4pif5*

quadruple mutant (*pifq*) becomes de-etiolated, morphologically mimicking wild-type Col-0 seedlings grown in the light with early signs of chloroplast biogenesis such as the disappearance of the prolamellar bodies, the development of rudimentary prothylakoid membranes, and the activation of PhANGs[15–17]. Besides PIFs, the repression of PhANGs in darkness requires two additional transcriptional regulators ETHYLENE-INSENSITIVE 3 (EIN3) and EIN3-LIKE 1 (EIL1), which transduce the signal of mechanical pressure when seedlings are buried under the ground[18]. Dark-grown *ein3/eil1* mutants, despite lacking obvious photomorphogenic phenotypes, show constitutive PhANG activation[18]. Corroborating the essential roles of PIFs and EIN3/EIL1 in repressing PhANGs, perturbing the stability of PIFs and/or EIN3/EIL1 in the classic *constitutive photomorphogenic/de-etiolated/fusca* (*cop/det/fus*) mutants also leads to activation of PhANGs in the absence of light[19,20]. CONSTITUTIVE PHOTOMORPHOGENIC 1 (COP1) is a zinc finger protein that either acts as an E3 ubiquitin ligase by itself or works together with SUPPRESSOR OF PHYTOCHROME A-105 (SPA) proteins as the substrate recognition subunit of DAMAGE-SPECIFIC DNA BINDING PROTEIN 1 (DDB1) and CULLIN 4 (CUL4) based E3 ubiquitin ligases (i.e., COP1-SPA-DDB1-CUL4 complexes)[21–23]. COP1 stabilizes PIF3 and EIN3/EIL1 by promoting the degradation of their cognitive E3 ubiquitin ligases, EIN3-BINDING F BOX PROTEIN 1 or 2[24]. In addition, the COP1-SPA complex interacts directly with PIF3 to block PIF3 phosphorylation by the protein kinase BRASSINOSTEROID INSENSITIVE 2 (BIN2) and the subsequent BIN2-dependent PIF3 degradation[25]. There are four SPA paralogs in *Arabidopsis*, named SPA1–4, of which SPA1 and SPA2 play prominent roles in repressing photomorphogenesis in the dark[26]. Moreover, DET1, together with COP10, constitutes the substrate recognition subunit of another DDB1-CUL4-based E3 ubiquitin ligase complex, COP10-DET1-DDB1-CUL4[27,28], which promotes PIF3 accumulation in the dark through direct DET1-PIF3 interaction[29]. PHYs and CRYs elicit photomorphogenesis by inhibiting the activity of the COP1-SPA complex in the light[30–33]. They also bind directly to PIFs to promote PIF phosphorylation, ubiquitylation, and proteasome-mediated degradation[12,13,24,34–36]. The light-dependent degradation of PIFs is considered a central mechanism to turn on the nuclear program for chloroplast biogenesis, particularly via the activation of PhANGs.

Light synchronizes PhAPG activation with PhANG expression via anterograde signaling[3]. Plastid genes are transcribed by two types of plastid RNA polymerases: a single-subunit, phage-type nuclear-encoded RNA polymerase (NEP) and a multi-subunit, bacterial-type plastid-encoded RNA polymerase (PEP)[37–39]. While the NEP preferentially transcribes housekeeping genes, including genes encoding the core PEP subunits, the PEP mainly transcribes PhAPGs[38]. Extensive biochemical and proteomics studies have demonstrated that a large fraction of the PEP is tightly associated with DNA and forms multi-subunit complexes[40,41]. The PEP complex comprises the prokaryotic α, β, β', β" core subunits surrounded by 12 PEP-associated proteins (PAPs) that are essential for the PEP activity[40,41]. Like the bacterial RNA polymerase, the PEP holoenzyme also requires a sigma factor (SIG) to specifically recognize promoter elements and initiate transcription[42]. There are 6 SIGs in *Arabidopsis*, named SIG1–6, which play unique and overlapping roles[42]. While the bacterial-like α, β, β', β" core subunits are encoded in the plastid genome by the *rpoA*, *rpoB*, *rpoC1*, and *rpoC2* genes, respectively, the 12 PAPs and 6 SIGs are encoded by the nuclear genome. We recently reported that PHY-mediated degradation of PIFs in the nucleus in monochromatic red light initiates an anterograde signaling pathway that triggers the assembly of the PEP into a 1000-kDa protein complex and transcription of PhAPGs[43,44]. These findings identify PIF degradation as a critical switch that synchronizes the activation of both PhANGs and PhAPGs.

The anterograde signaling mechanism downstream of PIFs to control the activity of the PEP remains elusive. The anterograde signal is likely encoded by a light-inducible gene repressed in darkness by

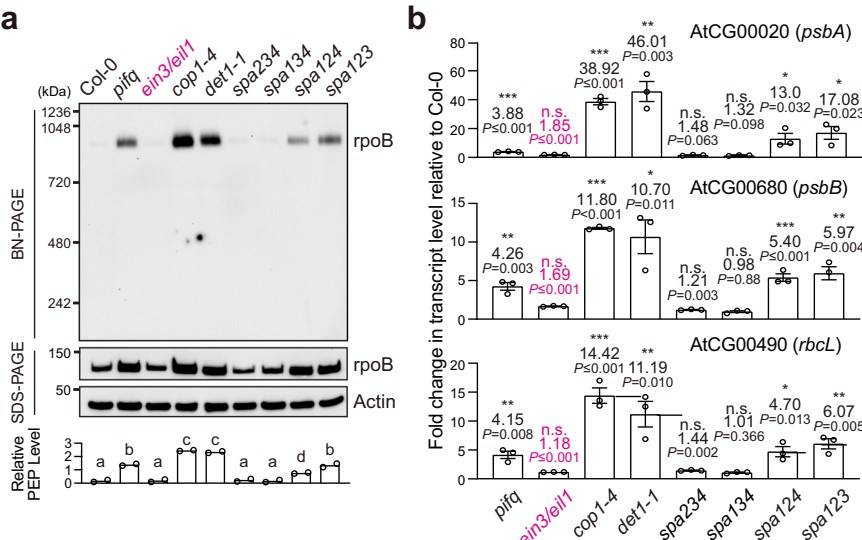

**Fig. 1 | The PEP is constitutively activated in all de-etiolated mutants except *ein3/eil1*. a** Immunoblots showing the levels of the PEP complex in 4-d-old dark-grown seedlings of Col-0, *pifq*, *ein3/eil1*, *cop1-4*, *det1-1*, spa*234*, spa*134*, spa*124*, and spa*123*. Total protein was isolated under either native or denaturing conditions and resolved via blue-native or sodium dodecyl sulfate polyacrylamide gel electrophoresis (BN-PAGE or SDS-PAGE) to assess the fraction of rpoB in the PEP complex or the amount of total rpoB by immunoblots, respectively. The bar graph shows the qualification of the relative levels of the PEP complex, which were calculated using the relative level of rpoB in the PEP complex (BN-PAGE) divided by the relative level of the total rpoB (SDS-PAGE) in each sample. Different letters denote statistically significant differences in the relative PEP levels among the genotypes

(ANOVA, Tukey's HSD, $P \le 0.05$, $n = 2$). **b** qRT-PCR results showing the fold changes in the steady-state transcript levels of *psbA*, *psbB*, and *rbcL* in 4-d-old dark-grown seedlings of *pifq*, *ein3/eil1*, *cop1-4*, *det1-1*, *spa234*, *spa134*, *spa124*, and *spa123* relative to their respective levels in Col-0 seedlings. Asterisks indicate a statistically significant and at least two-fold change in the transcript level in the mutant compared with that in Col-0 based on two-tailed Student's t-test (*$P \le 0.05$, **$P \le 0.01$, ***$P \le 0.001$); if the change was less than two-fold or not statistically significant, it is labeled as n.s. (not significant). Error bars represent the s.e. of three biological replicates, and the centers of the error bars represent the mean values. The source data underlying the immunoblots in **a** and the qRT-PCR analysis in **b** are provided in the Source Data file.

PIFs directly or indirectly. However, the main challenge is still to discern the gene encoding the signal molecule from among a large set of light-induced, PIF-repressed genes that encode proteins required for chloroplast transcription or other chloroplast functions. For example, despite the essential roles of PAPs and SIGs in PEP activity, whether these proteins could serve as anterograde signals for light-dependent PEP activation remains ambiguous. Also, PhAPGs are thought to be coupled with PhANGs. Because PhANGs are also repressed by PIFs and induced by light[15,16], the activation of PhANGs itself may constitute the anterograde signal. Supporting this idea, the ectopic expression of a light-harvesting complex protein is able to stimulate the formation of prothylakoid membranes in the dark[18]. Here, to dissect the anterograde signal, we first sought to determine whether light controls PhANGs and PhAPGs via hierarchical or parallel mechanisms. We characterized the relationship between the genome-wide induction of PhANGs and the activation of the PEP in *Arabidopsis* de-etiolated mutants that constitutively express PhANGs in darkness. These experiments surprisingly show that the genome-wide activation of PhANGs is insufficient to trigger PEP assembly in plastids, indicating that anterograde signaling controls plastid transcription via a separable pathway in parallel with the regulation of PhANGs. These experiments also allowed us to devise a strategy to distinguish anterograde signals from among essential components of the PEP holoenzyme. Together, our results further elucidate the framework of anterograde signaling that coordinates plastid transcription with the nuclear photosynthesis program during chloroplast biogenesis.

## Results

### PEP is constitutively activated in all de-etiolated mutants except *ein3/eil1*

To determine whether the activation of PhANGs in the nucleus plays a role in anterograde signaling, we examined PEP assembly and PhAPG activation in de-etiolated mutants that constitutively express PhANGs

even in darkness; these mutants included *pifq*[15,17], *cop1-4*[45], *det1-1*[46], four *spa* triple mutants (*spa234*, *spa134*, *spa124*, and *spa123*)[26], and *ein3/eil1*[47]. We asked whether the constitutive activation of PhANGs was sufficient to trigger PEP assembly and PhAPG induction in plastids. PEP assembly has not been examined in any of these mutants except *pifq*[43,44]. Notably, the criteria for "constitutive activation of PhANGs" were loosely defined at this point because the genome-wide expression of PhANGs had not been precisely characterized in most of these mutants. Thus, "constitutive activation of PhANGs" here merely means that at least some PhANGs were reported to be induced in the mutant in darkness.

We have previously shown that the PEP assembles into a 1000-kDa protein complex in *pifq* seedlings regardless of the light signal, leading to the constitutive expression of PhAPGs in the dark (Fig. 1a, b)[43,44]. To compare the levels of PEP assembly between genotypes, we performed immunoblots to quantify the "relative PEP level", which represents the relative level of rpoB from the PEP complex in total rpoB (Fig. 1). Consistent with the roles of COP1, DET1, and SPAs in stabilizing PIFs, the PEP was also constitutively assembled and activated in dark-grown *cop1-4*, *det1-1*, *spa124*, and *spa123* (Fig. 1a, b). The PEP was not activated in *spa234* or *spa134* (Fig. 1a, b), corroborating the idea that SPA1 and SPA2 exert major roles in maintaining skotomorphogenesis[26]. Together, these results suggest that COP1-SPA1-DDB1-CUL4 and COP1-SPA2-DDB1-CUL4 redundantly repress the anterograde signal that activates the PEP in darkness, likely by stabilizing PIFs. The COP1-SPA and the COP10-DET1 E3 ubiquitin ligases appeared to play non-redundant roles, as knocking out either one led to PEP activation (Fig. 1a, b). Because both PhANGs and PhAPGs were constitutively activated, these de-etiolated mutants did not provide new insights into the role of PhANGs in anterograde signaling.

A surprise came when we examined the *ein3/eil1* mutant. Strikingly, neither PEP assembly nor PhAPG expression was induced in dark-grown *ein3/eil1* seedlings (Fig. 1a, b), indicating that EIN3 and EIL1 do

not participate in anterograde signaling. Because it was suggested that EIN3 and EIL1 work collaboratively with PIF3 to repress PhANGs[18], the constitutive expression of PhANGs without the activation of PhAPGs in the *ein3/eil1* mutant might imply that the induction of PhANGs is insufficient to activate PhAPGs in the plastids.

## Only a subset of PhANGs are activated in *ein3/eil1*

Before we could reach a conclusion that the activation of PhANGs is insufficient to trigger PEP activation, we reasoned that we had to exclude the possibility that EIN3 and EIL1 could repress only a subset of PhANGs and PEP activation may require an activation of genome-wide PhANGs. To test this hypothesis, we decided to compare the global expression of PhANGs between dark-grown *pifq* and *ein3/eil1* mutants. The full set of PhANGs have not been specifically characterized in de-etiolated mutants. Most previous studies used representative PhANGs to infer the expression pattern of PhANGs genome wide, while a detailed transcriptomic analysis of precisely defined PhANGs is still lacking. For our analysis, we defined PhANGs as the 149 nuclear-encoded genes whose products directly participate in the photosynthetic reactions, such as light-harvesting (including chlorophyll biosynthesis), photosynthetic electron transport, ATP synthesis, and the Calvin cycle for $CO_2$ fixation (Supplementary Table 1)[3]. To identify possible distinct regulation between individual functional groups of PhANGs, we divided the PhANGs into three subcategories. Light harvesting for photosynthesis involves the light-harvesting complex that consists of chlorophyll molecules bound by LIGHT HARVESTING COMPLEX proteins (LHCs). All LHCs and chlorophyll biosynthetic enzymes are encoded by the nuclear genome; we named these two subcategories of PhANGs LHCs and CHLs (CHLorophyll biogenetic enzymes), respectively (Supplementary Table 1). We put the rest of the PhANGs, which encode components of photosystems I and II, electron transport, ATP synthase, and the Calvin cycle, into a third subcategory; because the components involved in these processes are encoded by both the nuclear and plastid genomes, we called this subcategory of PhANGs nuclear genes associated with Photosystems, Electron transport, ATP synthase, and the Calvin cycle (nPEACs) (Supplementary Table 1).

To assess the regulation of genome-wide PhANGs by PIFs and EIN3/EIL1, we reanalyzed published RNA-seq datasets on 4-d-old dark-grown *pifq* seedlings[29] and 3-d-old *ein3/eil1* seedlings[48] together with their respective Col-0 controls (Supplementary Table 2). To allow comparisons across different experimental settings, we normalized the transcript level of each gene against that of the commonly used control gene *PP2A* (AT1G13320) and then quantified the fold changes of the transcript levels of the PhANGs in the mutants relative to their respective Col-0 controls. We used a statistically significant two-fold change as the cutoff to define the genes that were differentially expressed in the mutants (Supplementary Data 1). Intriguingly, PhANGs exhibited conspicuously different expression patterns between *pifq* and *ein3/eil1*. In *pifq*, 119 of 149 PhANGs were upregulated compared with Col-0, these PIF-repressed genes comprised the majority of the genes in all three subcategories, LHC, CHL, and nPEAC, confirming that PIFs are the main repressors of chloroplast biogenesis by controlling about 80% of PhANGs (Fig. 2a, b and Supplementary Data 1). In contrast, only 10 PhANGs were upregulated in *ein3/eil1*; these EIN3/EIL1-repressed PhANGs were also repressed by the PIFs (Fig. 2a and Supplementary Data 1). Interestingly, nine of the ten PIF- and EIN3/EIL1-corepressed PhANGs were LHCs, especially *LHCB*s associated with photosystem II (Fig. 2b and Supplementary Data 1). We then plotted the expression changes of LHCs, CHLs, and nPEACs separately in *pifq* and *ein3/eil1* (Fig. 2c). The results demonstrated that whereas the majority of LHCs, CHLs, and nPEACs were upregulated in *pifq*, only LHCs were upregulated in *ein3/eil1* (Fig. 2b, c). It is important to note that gene expression can vary significantly in RNA-seq data generated from seedlings grown in different conditions or at different

developmental stages. A separate study showed that some nPEACs were also under the control of EIN3 and EIL1[18]. Nonetheless, both studies corroborate the idea that EIN3 and EIL1 repress only a subset of PhANGs, especially LHCs. We then analyzed the expression of PhANGs using published RNA-seq datasets generated from dark-grown 3-d-old *cop1-4* mutants[49], 4-d-old *det1-1* mutants[29], and 3-d-old *spa1234* quadruple mutants (*spaq*)[49]. Similar to *pifq*, all three subcategories of PhANGs were upregulated in *cop1-4*, *det1-1*, and *spaq* in the dark (Fig. 2c and Supplementary Data 1). To validate the RNA-seq results, we performed qRT-PCR experiments to measure the transcript levels of select LHCs and nPEACs in 4-d-old dark-grown Col-0, *pifq*, *ein3/eil1*, *cop1-4*, *det1-1*, *spa234*, *spa134*, *spa124*, and *spa123* seedlings. These results confirmed that COP1, SPA1/2, DET1, and PIFs repress both LHCs and nPEACs, whereas EIN3 and EIL1 repress only LHCs (Fig. 2d). Additionally, the three tested LHCs were activated at much lower levels in *ein3/eil1* than in the other de-etiolated mutants, suggesting that EIN3 and EIL1 play a relatively minor role compared with PIFs in repressing LHCs (Fig. 2d). Taken together, these results leave open the possibility that the lack of PEP activation in the *ein3/eil1* mutant could be due to the incomplete activation of PhANGs globally.

## Genome-wide activation of PhAPGs is insufficient to fully activate the PEP in *pifq*

To further examine the relationship between PhANG induction and PEP activation, we turned to a previously observed scenario in which PhAPGs stay repressed in *pifq* in early seedling development[43]. We have shown that neither PEP assembly nor PhAPG expression can be fully activated during early seedling development in 2-d-old dark-grown *pifq* seedlings (Fig. 3a, b)[43]. We asked whether the lack of PEP activation in 2-d-old *pifq* seedlings was also accompanied by incomplete global activation of PhANGs. To that end, we used published RNA-seq datasets from studies using 2-d-old[50] and 4-d-old[29] dark-grown *pifq* seedlings to analyze the expression of the 149 PhANGs (Supplementary Table 2). Intriguingly, 87 of the 119 upregulated PhANGs in 4-d-old dark-grown *pifq* seedlings were also upregulated in 2-d-old dark-grown *pifq* seedlings (Fig. 3c and Supplementary Data 2). All three subcategories of PhANGs were de-repressed in 2-d-old dark-grown *pifq* (Fig. 3d). Our qRT-PCR analysis on select LHCs and nPEACs confirmed the conclusion that PhANGs were activated in both 2-d and 4-d old *pifq* seedlings (Fig. 3e). These results thus argue against the hypothesis that the lack of PEP activation in 2-d-old *pifq* was due to the incomplete activation of PhANGs globally; instead, they provide evidence supporting that genome-wide PhANG activation is insufficient to trigger full PEP activation and PhAPG expression in plastids. These results also confirm that a developmental signal exists during early seedling development to specifically repress the anterograde signal for PEP activation independently from the regulation of PhANGs.

## Ectopic expression of *LHCB*s does not trigger PEP activation

It was previously shown that ectopically expressing *LHCB1.1* or *LHCB2.1* triggers the transition from etioplasts to chloroplasts in the dark[18]. Plastids in dark-grown seedlings overexpressing *LHCB1.1* or *LHCB2.1* (designated *LHCB1.1ox* and *LHCB2.1ox*, respectively) lacked prolamellar bodies, and instead, developed prothylakoid membranes, indicative of the initiation of chloroplast biogenesis. However, neither PEP assembly nor PhAPG expression was induced in these lines in the dark (Fig. 4a, b), supporting the idea that the induction of PhANGs alone is not sufficient to trigger PEP activation. These results also imply that the activation of PhAPGs does not necessarily correlate with the restructuring of the internal membrane system during the early etioplast-to-chloroplast transition.

## Multiple PIFs redundantly repress the anterograde signal

Photomorphogenesis is repressed in the dark primarily by four PIFs, PIF1, PIF3, PIF4, and PIF5[15,17]. We asked whether the anterograde signal

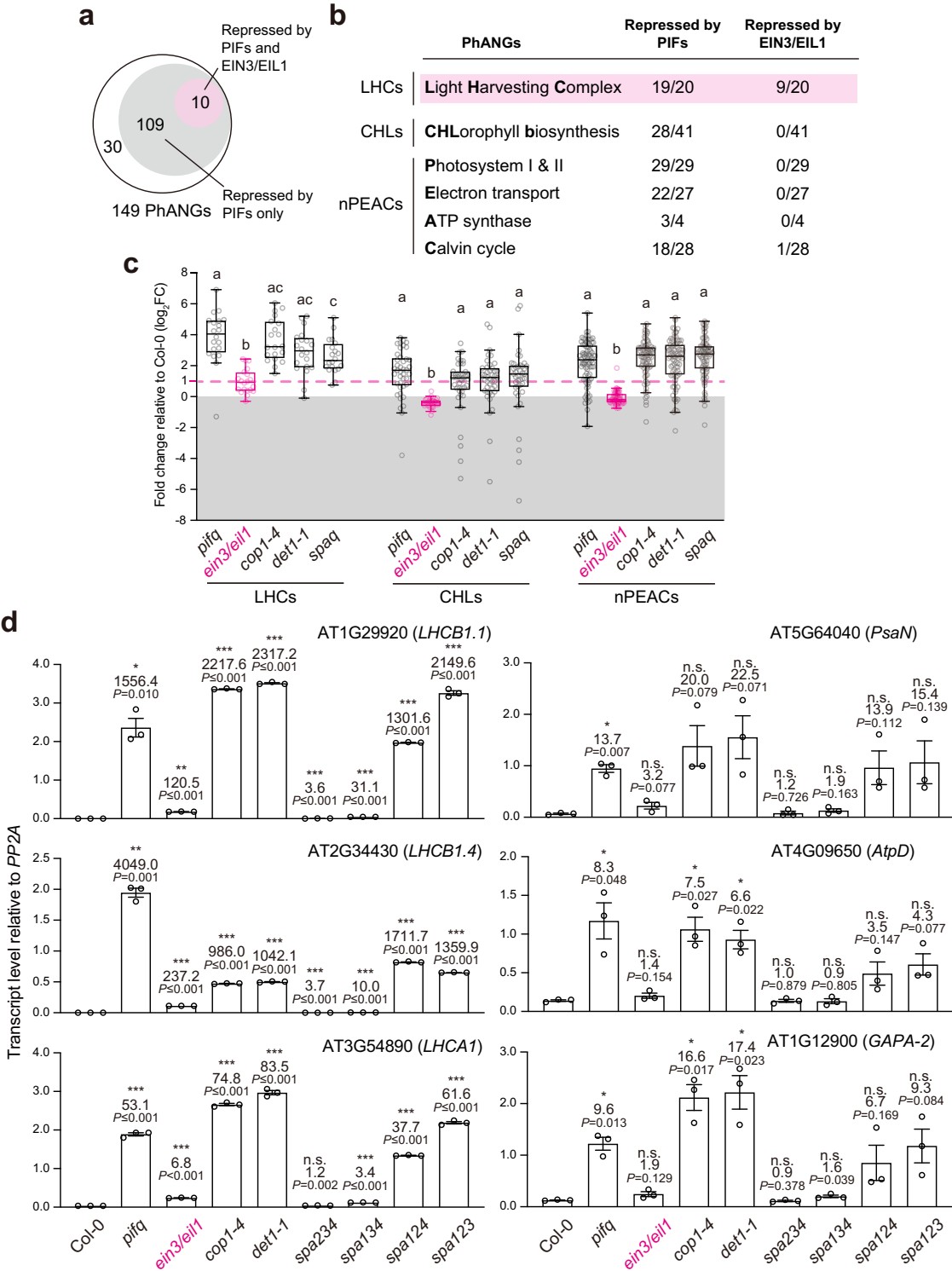

**Fig. 2 | Only a subset of PhANGs are constitutively activated in *ein3/eil1*. a** Venn diagram showing PhANGs repressed by PIFs and EIN3/EIL1 in dark-grown seedlings. **b** Table showing the number of PIF-repressed and EIN3/EIL1-repressed genes in the PhANG subcategories. **c** Box-and-whisker plots showing the fold changes in LHCs, CHLs, and nPEACs in *pifq*, *ein3/eil1*, *cop1-4*, *det1-1*, and *spaq* compared with Col-0 seedlings. Each data point represents a gene; data points above the dotted magenta line are statistically significantly upregulated in the mutants by at least two-fold. The lower and upper bounds of the box represent the 25th and 75th percentile respectively; the center bar represents the median. The lower and upper end of the whiskers represent minimum and maximum values respectively. Dots outside of the whiskers represent outliers based on Tukey's fences. Different letters denote statistically significant differences in gene expression among the mutants within each subcategory (ANOVA, Tukey's HSD, *P* ≤ 0.05; *n* = 3 biological

replicates for *pifq*, *cop1-4*, *det1-1*, and *spaq*, and *n* = 2 biological replicates for *ein3/eil1*). **d** qRT-PCR analysis of the steady-state transcript levels of representative LHCs (*LHCB1.1*, *LHCB1.4*, and *LHCA1*) and nPEACs (*PsaN*, *AtpD*, and *GAPA-2*) in 4-d-old dark-grown seedlings of Col-0, *pifq*, *ein3/eil1*, *cop1-4*, *det1-1*, *spa234*, *spa134*, *spa124*, and *spa123*. The transcript levels were calculated relative to those of *PP2A*. The fold changes between the transcript levels in the mutants and Col-0 are shown. Asterisks indicate statistically significant differences between the transcript levels of the mutant and Col-0 based on two-tailed Student's *t*-test (*\*P* ≤ 0.05, *\*\*P* ≤ 0.01, *\*\*\*P* ≤ 0.001). If the change was less than two-fold or not statistically significant, it is labeled as n.s. (not significant). Error bars represent the s.e. of three biological replicates, and the centers of the error bars represent the mean values. The source data underlying the statistical analysis in **c** and qRT-PCR analysis in **d** are provided in the Source Data file.

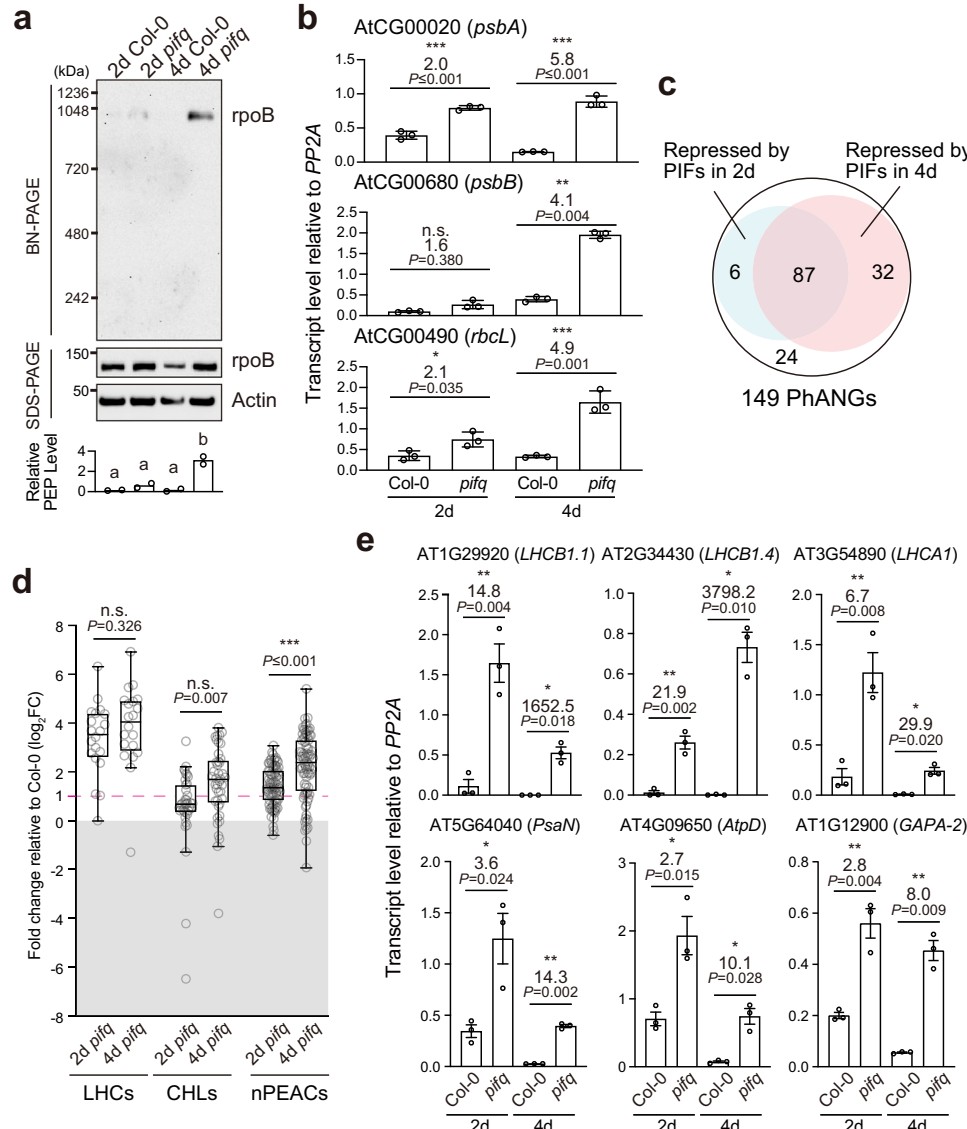

**Fig. 3 | Genome-wide activation of PhANGs is insufficient to fully activate the PEP in *pifq*. a** Immunoblots showing the levels of the PEP complex and its core component rpoB in 2-d-old and 4-d-old Col-0 and *pifq* seedlings grown in the dark. Total protein was isolated under either native or denaturing conditions and resolved via BN-PAGE or SDS-PAGE to assess the fraction of rpoB in the PEP complex or the amount of total rpoB by immunoblots, respectively. The bar graph shows the qualification of the relative levels of the PEP complex, which were calculated using the relative level of rpoB in the PEP complex (BN-PAGE) divided by the relative level of the total rpoB (SDS-PAGE) in each sample. Different letters denote statistically significant differences in the relative PEP levels among the genotypes (ANOVA, Tukey's HSD, $P \leq 0.05$, $n = 2$). **b** qRT-PCR analysis of the steady-state transcript levels of *psbA*, *psbB*, and *rbcL* in 2-d-old and 4-d-old Col-0 and *pifq* seedlings grown in the dark. The transcript levels were calculated relative to those of *PP2A*. The fold changes between the transcript levels in the mutants and Col-0 are shown. Asterisks indicate statistically significant differences in the transcript levels between the mutant and Col-0 based on two-tailed Student's *t*-test (*$P \leq 0.05$, **$P \leq 0.01$, ***$P \leq 0.001$). If the change was less than two-fold or not statistically significant, it is labeled as n.s. (not significant). Error bars represent the s.e. of three biological replicates, and the centers of the error bars represent the mean values. **c** Venn diagram showing PIF-repressed PhANGs in 2-d-old and 4-d-old dark-grown

seedlings. **d** Box-and-whisker plots showing the fold changes in LHCs, CHLs, and nPEACs in 2-d-old and 4-d-old dark-grown *pifq* seedlings compared with the Col-0 controls. Each data point represents a gene; data points above the dotted magenta line are statistically significantly upregulated in the mutants by at least two-fold. The lower and upper bounds of the box represent the 25th and 75th percentile respectively; the center bar represents the median. The lower and upper end of the whiskers represent minimum and maximum values respectively. Dots outside of the whiskers represent outliers based on Tukey's fences. Asterisks indicate statistically significant differences between the 2-d and 4-d samples based on two-tailed Student's *t*-test (***$P \leq 0.001$, $n = 6$ biological replicates for 2-d-old dark-grown *pifq*, and $n = 3$ biological replicates for 4-d-old dark-grown *pifq*). n.s. indicates no significant change. **e** qRT-PCR analysis of the steady-state transcript levels of representative LHCs (*LHCB1.1*, *LHCB1.4*, and *LHCA1*) and nPEACs (*PsaN*, *AtpD*, and *GAPA-2*) in 2-d-old and 4-d-old dark-grown Col-0 and *pifq* seedlings. The transcript levels were calculated relative to those of *PP2A*. The fold changes between the mutants and Col-0 are shown. Asterisks indicate statistically significant differences between the mutant and Col-0 based on two-tailed Student's *t*-test (*$P \leq 0.05$, **$P \leq 0.01$). Error bars represent the s.e. of three biological replicates, and the centers of the error bars represent the mean values. The source data underlying the immunoblots in **a** and the qRT-PCR analysis in **b** and **e** are provided in the Source Data file.

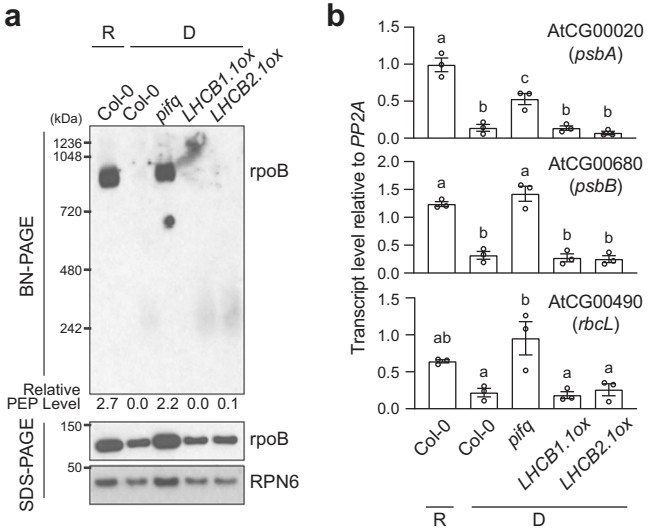

**Fig. 4 | Ectopic expression of *LHCB*s does not activate the PEP. a** Immunoblots showing the levels of the PEP complex, as well as its core component rpoB, in 4-d-old Col-0 seedlings grown in 10 μmol m⁻²sec⁻¹ R light and 4-d-old Col-0, *pifq*, *LHCB1.1ox*, and *LHCB2.1ox* seedlings grown in the dark. Total protein was isolated under either native or denaturing conditions and resolved via BN-PAGE or SDS-PAGE to assess the fraction of rpoB in the PEP complex or the amount of total rpoB by immunoblots, respectively. The numbers below the immunoblots represent the relative PEP levels, which were calculated using the relative level of rpoB in the PEP complex (BN-PAGE) divided by the relative level of denatured rpoB (SDS-PAGE) in each sample. **b** qRT-PCR analysis of the steady-state transcript levels of select PhAPGs, *psbA*, *psbB*, and *rbcL*, in 4-d-old Col-0 seedlings grown in 10 μmol m⁻²sec⁻¹ R light and 4-d-old Col-0, *pifq*, *LHCB1.1ox*, and *LHCB2.1ox* seedlings grown in the dark. The transcript levels were calculated relative to those of *PP2A*. Error bars represent the s.e. of three biological replicates, and the centers of the error bars represent the mean values. Samples labeled with different letters exhibited statistically significant differences (ANOVA, Tukey's HSD, $P \leq 0.05$, $n = 3$). The source data underlying the immunoblots in **a** and the qRT-PCR analysis in **b** are provided in the Source Data file.

is repressed by individual PIFs or redundantly by multiple PIFs. To assess the role of individual PIFs in PEP inhibition, we examined PEP assembly and the expression of PhAPGs in *pif1*, *pif3*, *pif4*, and *pif5* single mutants, as well as *pif345*, *pif145*, *pif135*, and *pif134* triple mutants, which leave one of the four PIFs intact. Knocking out the four PIFs individually did not have a dramatic impact on PEP assembly, although the *pif1* mutant showed a minor but significant increase in the level of the PEP complex compared with Col-0 (Fig. 5a). The individual *pif* mutants also did not show a significant increase in the transcript levels of select PhAPGs compared with Col-0 (Fig. 5b). These results indicate that PEP inhibition is imposed not by a particular PIF but rather redundantly by more than one PIF. Among the triple mutants, all exhibited a significant decrease in the level of the PEP complex compared with *pifq* (Fig. 5a), none of the four *pif* triple mutants could fully activate the three representative PhAPGs as strongly as could *pifq* (Fig. 5b), supporting the idea that the four PIFs repress the anterograde signal redundantly. The single *pif* mutants were also not effective in activating PhANGs, and none of the triple *pif* mutants could activate PhANGs to the same extent as *pifq* could (Fig. 5c). Therefore, the four PIFs act redundantly to repress both PhANGs and PhAPGs.

### Repression of the anterograde signal requires PIF3's activator activity

PIFs are transcriptional activators that promote skotomorphogenesis mainly by activating target genes, although PIFs can also repress gene expression[14]. Therefore, PIFs could repress the anterograde signal

either directly or indirectly via PIF-induced genes. We recently identified a 24-amino-acid transcription activation domain (AD) in PIF3 that consists of a ΦxxΦΦ activator motif, where Φ indicates a bulky hydrophobic residue and x is any other amino acid, flanked by acidic residues[11]. A PIF3mAD mutant, which substitutes the ΦxxΦΦ activator motif with five alanines, abolishes the activator activity of PIF3[11]. To determine whether the transactivation activity of PIF3 is required for repressing the anterograde signal, we examined PEP inhibition in dark-grown *PIF3/pifq* and *PIF3mAD/pifq* lines that express HA-YFP-PIF3 or HA-YFP-PIF3mAD, respectively, in the *pifq* background. As expected, the *PIF3/pifq* lines largely complemented the *pifq* mutant and could repress PEP assembly and PhAPG expression in darkness (Fig. 6a). In contrast, the *PIF3mAD/pifq* lines only partially repressed PEP assembly and PhAPGs remained fully activated as in *pifq* (Fig. 6a, b). These results thus suggest that PIF3 repress the anterograde signal indirectly via a PIF3-induced gene product. Given that PIFs play similar roles in repressing the anterograde signal (Fig. 5) and that all four PIFs have the same type of AD[11], it is likely that other PIFs also repress the anterograde signal indirectly through their activator activity.

### PIF-repressed sigma factors are anterograde signals

We next asked whether the nuclear-encoded components of the PEP – i.e., 12 PAPs and 6 SIGs – could serve as anterograde signals downstream of PIFs. Here we used the criteria that an anterograde signal must be encoded by a PIF-repressed gene that is also repressed by the developmental signal in early seedling development. To that end, we asked whether the expression patterns of any of these components match the pattern of PEP activation in *pifq* and *ein3eil1*; in other words, we examined whether any of the PEP components were induced in 4-d-old *pifq* but remained repressed in 4-d-old *ein3eil1* and 2-d-old *pifq*. Intriguingly, none of the 12 *PAP*s were upregulated in *pifq* or the other de-etiolated mutants in the dark (Supplementary Fig. 1), suggesting that the PAPs are not regulated by PIFs at the transcript level. In contrast, four *SIG*s, including *SIG1*, *SIG3*, *SIG5*, and *SIG6*, were induced in 4-d-old *pifq*; we therefore refer to them as the PIF-repressed *SIG*s. More strikingly, these four PIF-repressed *SIG*s were not activated in 4-d-old *ein3/eil1* and 2-d-old *pifq* (Fig. 7a), matching the pattern of PEP activation.

SIGs are an essential component of the PEP holoenzyme for promoter recognition and transcription initiation. However, bacterial SIGs are not required for the assembly of the core subunits into the RNA polymerase complex[51]. We then tested whether the accumulation of the PEP complex was affected in *sig* mutants during chloroplast biogenesis in seedling development. To that end, we first examined PEP complex formation in Col-0 at different days after seed imbibition. Interestingly, the PEP complex and PhAPGs were repressed in 1-d-old seedlings, consistent with the idea that a developmental signal exists to inhibit the activation of plastid gene expression during early seedling establishment. The PEP complex became visible in 2-to-4 day old seedlings. Among the six SIGs, SIG2 and SIG6 are major general SIGs, as *sig2-2* and *sig6-1* mutants exhibit pale green phenotypes[52]. We then examined the PEP complex in the *sig6-1* mutant because *sig6-1* was the only mutant showing a visible phenotype among the mutants of the four PIF-repressed *SIG*s. The relative PEP level was only reduced significantly in 2-d-old *sig6-1* but remained the same compared with Col-0 in 3-d and 4-d old *sig6-1* (Fig. 7b). These results suggest that SIG6 is not required for the PEP complex, although the accumulation of the PEP could be influenced in the *sig6-1* mutant likely as a secondary effect. We then examined the PEP complex in 2-d-old seedlings of the other *sig* mutants. Only *sig2-2* – the other *sig* mutant with a visible phenotype – showed a reduction in the PEP complex (Fig. 7c). Therefore, we conclude that plastid SIGs are not required for the assembly of the PEP complex, despite their essential roles in promoter recognition and transcription initiation.

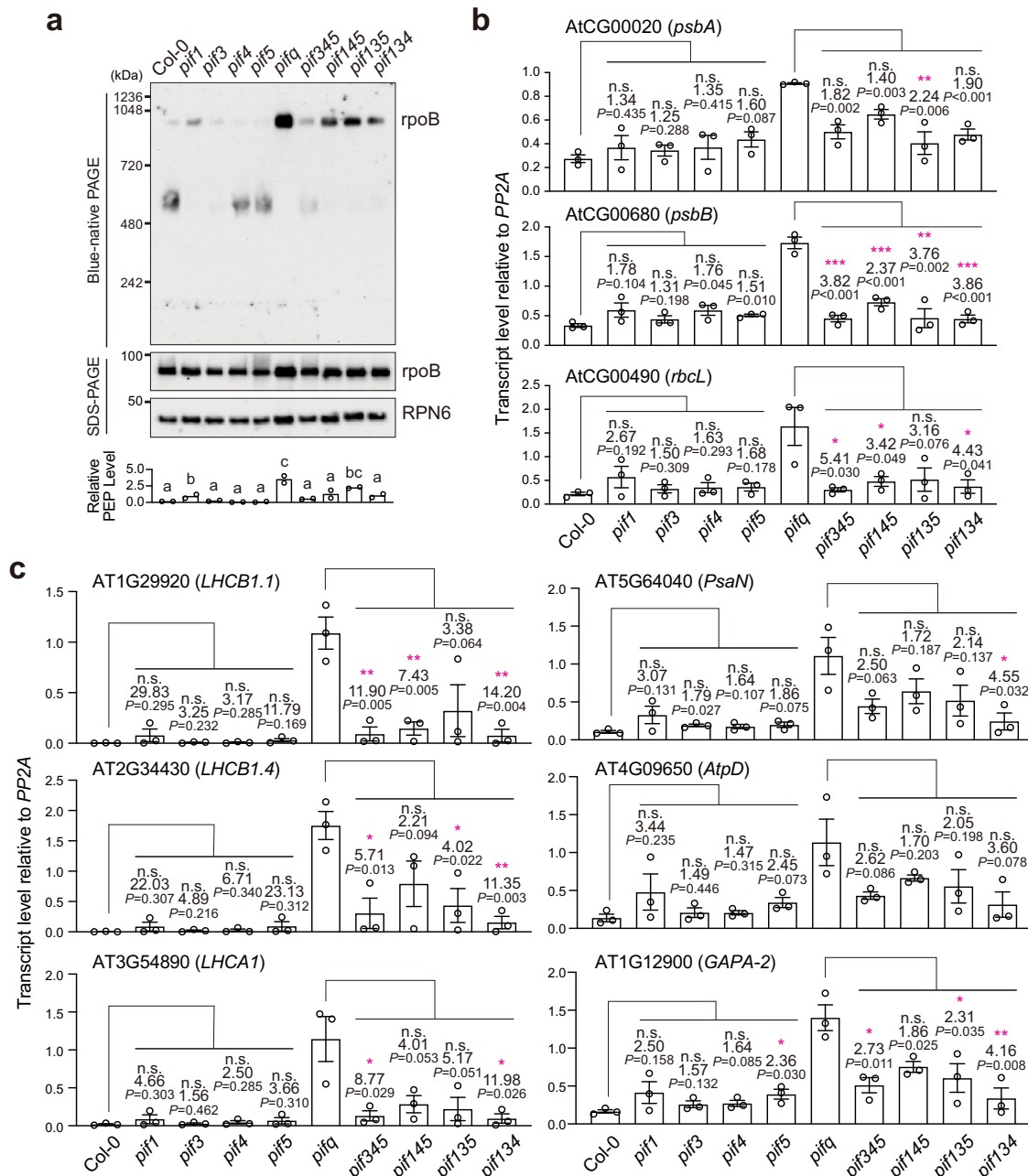

**Fig. 5 | PEP activation is repressed redundantly by multiple PIFs.**
**a** Immunoblots showing the levels of the PEP complex and its core component rpoB in 4-d-old Col-0, *pif1*, *pif3*, *pif4*, *pif5*, *pifq*, *pif345*, *pif145*, *pif135*, and *pif134* seedlings grown in the dark. Total protein was isolated under either native or denaturing conditions and resolved via BN-PAGE or SDS-PAGE to assess the fraction of rpoB in the PEP complex or the amount of total rpoB by immunoblots, respectively. RPN6 was used as a loading control. The graph below the immunoblots shows the relative PEP levels, which were estimated using the relative level of rpoB in the PEP complex (BN-PAGE) divided by the relative level of denatured rpoB (SDS-PAGE) in each sample. Different letters denote statistically significant differences in relative PEP level either between the *pif* single mutants and Col-0 or between the *pif* triple mutants and *pifq* (ANOVA, Tukey's HSD, $P \leq 0.05$, $n = 2$).

**b** qRT-PCR analysis of the steady-state transcript levels of select PhAPGs, including *psbA*, *psbB*, and *rbcL*, in the 4-d-old dark-grown seedlings shown in **a**. **c** qRT-PCR analysis of the steady-state transcript levels of select PhANGs in the 4-d-old dark-grown seedlings described in **a**. For **b** and **c**, asterisks indicate statistically significant differences in the transcript levels either between the single *pif* mutants and Col-0 or between the *pif* triple mutants and *pifq* based on two-tailed Student's *t*-test (*$P \leq 0.05$, **$P \leq 0.01$, ***$P \leq 0.001$). If the change was less than two-fold or not statistically significant, it is labeled as n.s. (not significant). Error bars represent the s.e. of three biological replicates, and the centers of the error bars represent the mean values. The source data underlying the immunoblots in **a** and the qRT-PCR analysis in **b** and **c** are provided in the Source Data file.

## PIF-repressed sigma factors are rapidly induced by light

In addition to being PIF-repressed, anterograde signals triggering PEP activation should also be induced by light. The light-dependent inhibition of the stability and activity of PIFs is a master nuclear switch that activates both PhANGs and PhAPGs. PHYA and CRYs, which sense far-red and blue light, respectively, regulate PIFs in a similar manner as PHYB[12,53]. PhANGs can be activated by monochromatic light via PHYA and PHYB in red light[15,16], by PHYA in far-red light[54,55], and by CRYs in blue light[56]. To test whether PHYA and CRYs can also individually activate PhAPGs, we examined the assembly of the PEP and the expression of representative PhAPGs in monochromatic far-red and blue light. Growing Col-0 seedlings in monochromatic far-red light was

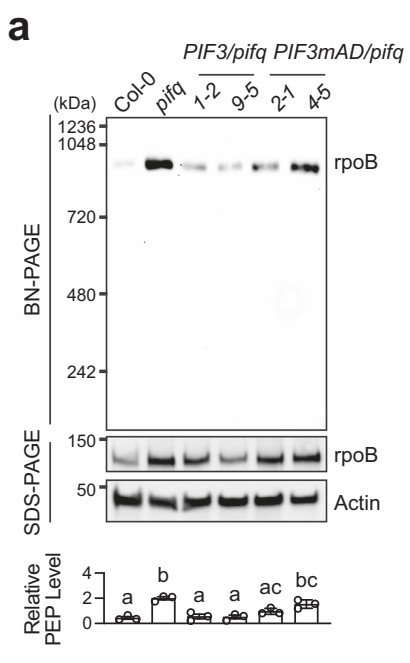

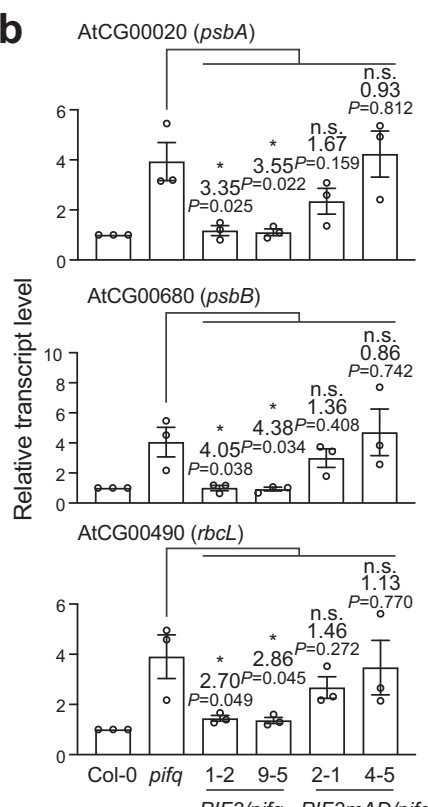

**Fig. 6 | PEP inhibition requires PIF3's transactivation activity. a** Immunoblots showing the levels of the PEP complex and its core component rpoB in 4-d-old Col-0, *pifq*, *PIF3/pifq* (lines 1-2 and 9-5), and *PIF3mAD/pifq* (lines 2-1 and 4-5) seedlings grown in the dark. Total protein was isolated under either native or denaturing conditions and resolved via BN-PAGE or SDS-PAGE to assess the fraction of rpoB in the PEP complex or the amount of total rpoB by immunoblots, respectively. Actin was used as a loading control. The graph below the immunoblots shows the relative PEP levels, which were estimated using the relative level of rpoB in the PEP complex (BN-PAGE) divided by the relative level of denatured rpoB (SDS-PAGE) in each sample. Different letters denote statistically significant differences in the relative PEP levels (ANOVA, Tukey's HSD, $P \leq 0.05$, $n = 3$). Error bars represent the s.e. of three biological replicates, and the centers of the error bars represent the mean values. **b** qRT-PCR analysis of the steady-state transcript levels of select PhAPGs, including *psbA*, *psbB*, and *rbcL*, in the 4-d-old dark-grown seedlings shown in **a**. The transcript levels were calculated relative to those of *PP2A*. Error bars represent the s.e. of three biological replicates, and the centers of the error bars represent the mean values. Asterisks indicate statistically significant differences between the indicated transgenic lines and *pifq* based on two-tailed Student's *t*-test (*$P \leq 0.05$). If the change was less than two-fold or not statistically significant, it is labeled as n.s. (not significant). The source data underlying the immunoblots in **a** and the qRT-PCR analysis in **b** are provided in the Source Data file.

sufficient to trigger both PEP assembly and PhAPG activation (Fig. 8a, b). The transcript levels of *psbA*, *psbB*, and *rbcL* were significantly higher in far-red than in red light (Fig. 8b), suggesting that either the plastid photosynthesis genes were more actively transcribed or their transcripts became more stable in far-red light. Both the far-red-mediated PEP assembly and PhAPG expression were blocked in *phyA-211* (Fig. 8a, b), indicating that PHYA is responsible for PEP activation in far-red light. Monochromatic blue light was equally effective in eliciting PEP assembly and PhAPG activation (Fig. 8a, b). The blue light responses were abolished in the *cry1/cry2* mutant, indicating that CRYs play an essential role in activating the PEP in blue light. PHYA can also detect blue light[57], but PEP assembly and PhAPG expression could still be induced by blue light in *phyA-211* (Fig. 8a, b), indicating that PHYA does not play a major role in regulating PhAPGs in blue light. Together, these results support the conclusion that PHYA and CRYs can individually initiate anterograde signaling to activate the PEP in monochromatic far-red and blue light, respectively. The PEP assembly was mostly unaffected in the *sig* mutants in far-red and blue light (Supplementary Fig. 2), supporting the conclusion that SIGs are not required for PEP assembly despite being an essential component for the activity of PEP holoenzyme.

We then analyzed the expression of 12 *PAP*s and 6 *SIG*s during the dark-to-light transition using a published RNA-seq dataset on 3-d-old dark-grown Col-0 seedlings treated with 1 or 3 h of red, far-red, blue, or white light[58]. Based on their expression patterns, the 18 nuclear-encoded PEP components were grouped into two clusters – a light-upregulated and a light-downregulated cluster (Fig. 8c). The majority of the *PAP*s were expressed at relatively high levels in the dark and their expression was downregulated during the dark-to-light transition. Intriguingly, all four PIF-repressed *SIG*s – *SIG1*, *SIG3*, *SIG5*, and *SIG6* – belonged to the light-induced cluster, which included all six *SIG*s and three *PAP*s, *PAP2*, *PAP10*, and *PAP11*. The cluster of light-inducible PEP components were highly expressed in far-red light (also in white light), consistent with the elevated expression of PhAPGs in far-red light (Fig. 8b,c). These results further support the role of the four PIF-repressed SIGs as anterograde signals. The other light-inducible PEP components could also contribute to anterograde signaling; the fact that their expression was not significantly altered in *pifq* implies that anterograde signaling involves PIF-independent mechanisms.

## Discussion

The nuclear control of plastid transcription is pivotal for the coordination of nuclear and plastid gene expression during chloroplast biogenesis[3,4]. We recently revealed that red-light-dependent photoinhibition of PIFs by PHYB in the nucleus triggers the assembly of the PEP and PhAPG transcription in plastids (Fig. 9)[3,43,44]. It is conceivable

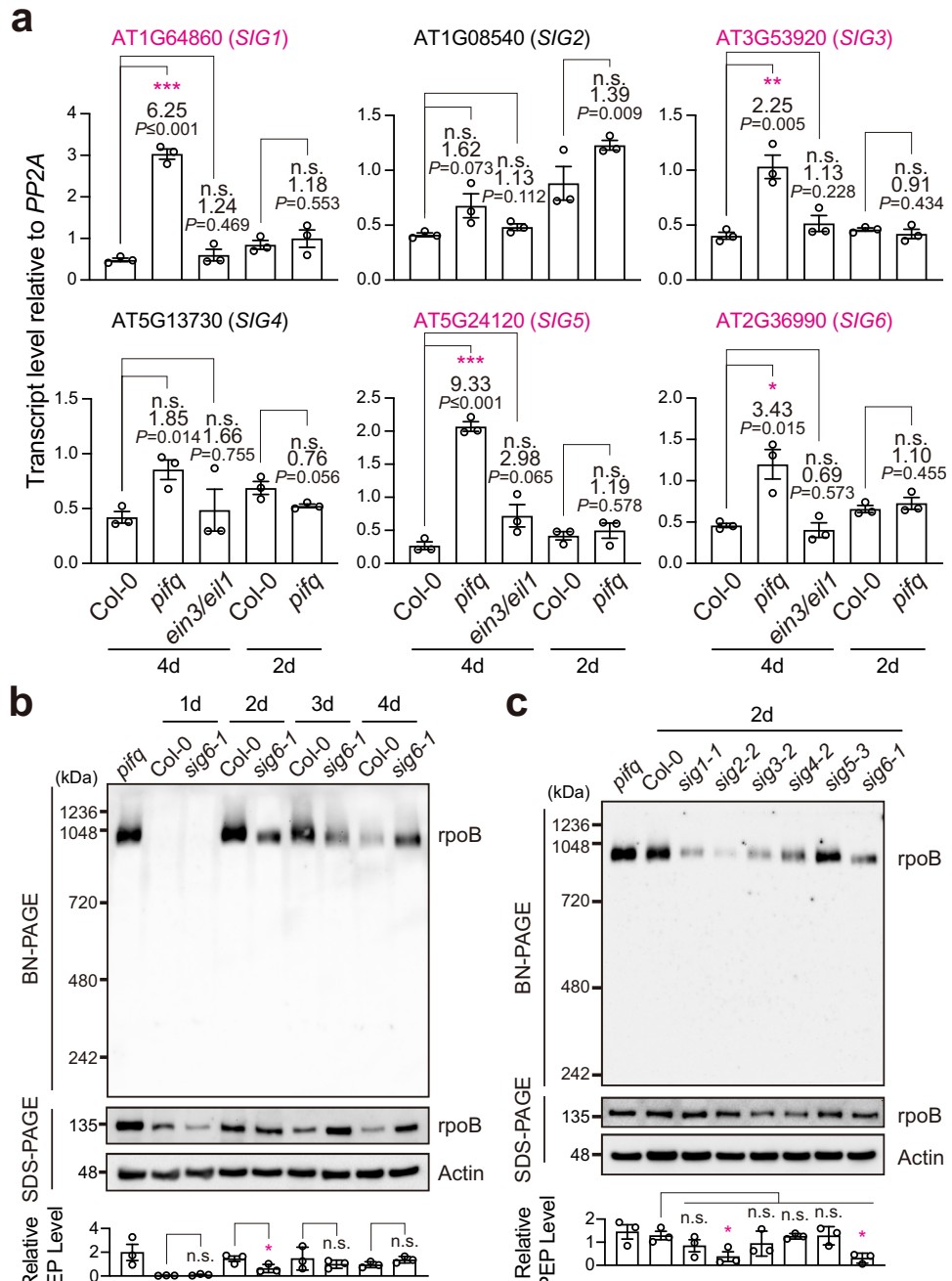

**Fig. 7 | PEP activation correlates with the expression of PIF-repressed sigma factors. a** qRT-PCR analysis of the steady-state transcript levels of *SIG1-6* in dark-grown 4-d-old Col-0, *pifq*, and *ein3/eil1*, as well as 2-d-old Col-0 and *pifq* seedlings. The transcript levels were calculated relative to those of *PP2A*. The fold changes between the transcript levels in the mutants and their respective Col-0 controls are shown. Asterisks indicate statistically significant differences in the transcript level compared with that of Col-0 based on two-tailed Student's *t*-test (*$P \le 0.05$, **$P \le 0.01$, ***$P \le 0.001$). If the change was less than two-fold or not statistically significant, it is labeled as n.s. (not significant). Error bars represent the s.e. of three biological replicates, and the centers of the error bars represent the mean values. **b** Immunoblots showing the levels of the PEP complex and the core PEP component rpoB in Col-0, and *sig6-1* seedlings grown in 10 μmol m$^{-2}$sec$^{-1}$ R light for 1 to 4 days after imbibition. **c** Immunoblots showing the levels of the PEP complex and the core PEP component rpoB in 2-d-old Col-0, *sig1-1*, *sig2-2*, *sig3-2*, *sig4-2*, *sig5-3*,

and *sig6-1* seedlings grown in 10 μmol m$^{-2}$sec$^{-1}$ R light. For **b** and **c**, 4-d-old *pifq* seedlings grown in the dark were used as a control. Total protein was isolated under either native or denaturing conditions and resolved BN-PAGE or SDS-PAGE to assess the fraction of rpoB in the PEP complex or the amount of total rpoB by immunoblots, respectively. Actin was used as a loading control. The graph below the immunoblots shows the relative PEP levels, which were estimated using the relative level of rpoB in the PEP complex (BN-PAGE) divided by the relative level of denatured rpoB (SDS-PAGE) in each sample. Asterisks indicate statistically significant differences with Col-0 in triplicates based on two-tailed Student's *t*-test (*$P \le 0.05$). Error bars represent the s.e. of three biological replicates, and the centers of the error bars represent the mean values. The source data underlying the qRT-PCR analysis in **a** and the immunoblots in **b** and **c** are provided in the Source Data file.

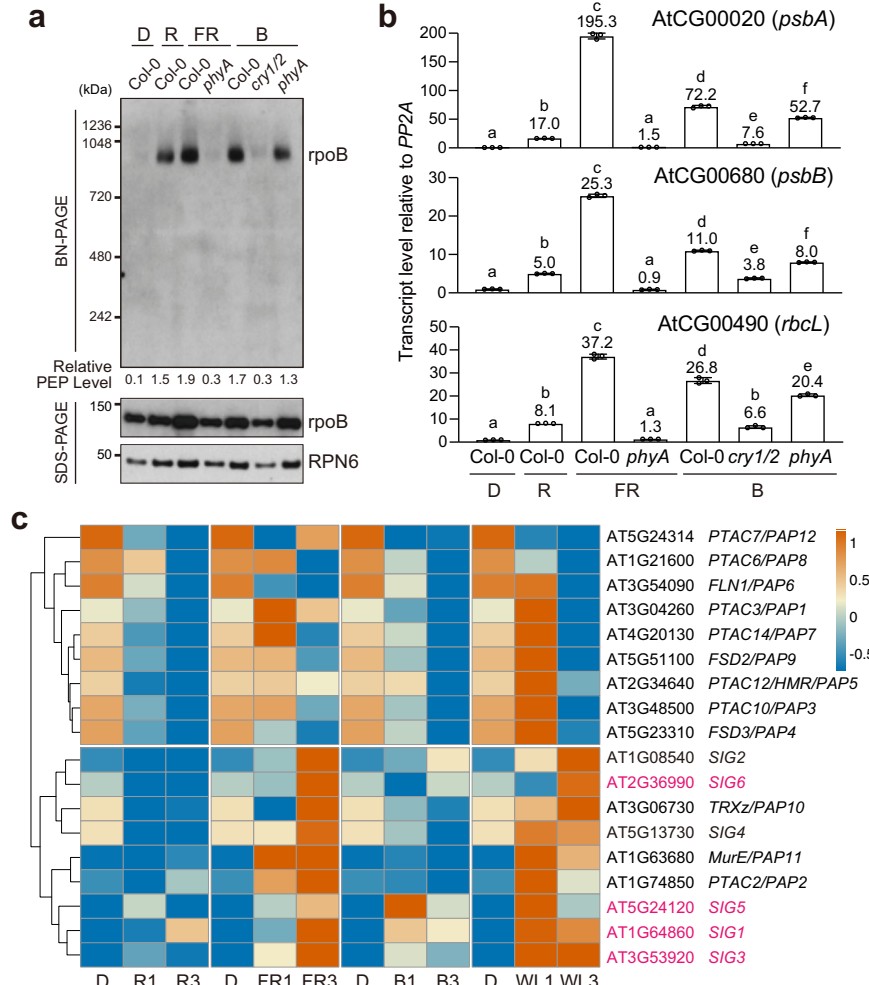

**Fig. 8 | PIF-repressed sigma factors are rapidly induced by light. a** Immunoblots showing the levels of the PEP complex and the PEP component rpoB in 4-d-old Col-0, *phyA-211* (*phyA*), and *cry1/cry2* (*cry1/2*) seedlings grown in the dark (D) or in 10 μmol m$^{-2}$s$^{-1}$ monochromatic red (R), far-red (FR), or blue (B) light. Total protein was isolated under either native or denaturing conditions and resolved via BN-PAGE or SDS-PAGE to assess the fraction of rpoB in the PEP complex or the amount of total rpoB by immunoblots, respectively. RPN6 was used as a loading control. The numbers below the immunoblots represent the relative PEP levels, which were estimated using the relative level of rpoB in the PEP complex (BN-PAGE) divided by the relative level of denatured rpoB (SDS-PAGE) in each sample. **b** qRT-PCR analysis of the steady-state transcript levels of *psbA*, *psbB*, and *rbcL* in 4-d-old Col-0, *phyA-*

*211* (*phyA*), and *cry1/cry2* (*cry1/2*) seedlings grown in the dark or the light conditions described in **a**. The transcript levels were calculated relative to those of *PP2A*. The numbers above the columns indicate the fold changes compared with Col-0 in the dark. Error bars represent the s.e. of three biological replicates, and the centers of the error bars represent the mean values. Samples labeled with different letters exhibited statistically significant differences (ANOVA, Tukey's HSD, $P \leq 0.05$, $n = 3$). **c** Heatmap showing the relative expression of 12 *PAP*s and 6 *SIG*s in 3-d-old Col-0 during the transition from dark to red (R), far-red (FR), blue (B), and while light (WL) for 1 or 3 h. The source data underlying the immunoblots in **a**, qRT-PCR analysis in **b**, and the heatmap in **c** are provided in the Source Data file.

that, instead of all PIF-regulated plastid proteins being employed as signaling molecules, likely only some act as regulators or anterograde signals to confer the nuclear control of plastid gene expression in a spatial and temporal manner. For decades, the main challenge has been to identify the anterograde signals from among the battery of nuclear-encoded components required for plastid transcription or other essential chloroplast functions, such as photosynthesis[5]. In this study, we demonstrated that, although PIFs repress both PhANGs and PhAPGs, neither selective nor genome-wide induction of PhANGs is sufficient to activate PhAPGs. Therefore, PhANGs themselves do not constitute the anterograde signal; instead, anterograde signaling activates the PEP via a separate mechanism in parallel with the regulation of PhANGs. Based on the pattern of PEP activation in de-etiolated mutants, we devised a strategy – using transcriptomic analysis of the *pifq* mutant at two seedling developmental stages in combination with the analysis of early light-responsive genes in Col-0 – to distinguish anterograde signals from essential components of the

PEP. Among the 18 nuclear-encoded components of the PEP holoenzyme, this approach identified four light-inducible, PIF-repressed SIGs, including SIG1, SIG3, SIG5, and SIG6, as anterograde signals downstream of PIFs to control PEP activation (Fig. 9). Our results also predict the existence of a developmental mechanism that represses the anterograde signal during early seedling establishment (Fig. 9). Moreover, we show that far-red and blue light could individually turn on anterograde signaling via PHYA and CRYs, respectively (Fig. 9). Together, these results further elucidate the framework of light signaling that coordinates nuclear and plastid photosynthesis genes, in which PHYs and CRYs synchronize plastid transcription via PIF-repressed SIGs as anterograde signals in parallel with the regulation of nuclear photosynthesis genes (Fig. 9).

The provocative conclusion of a genetically uncoupled regulation of PhANGs and PhAPGs extends the conventional view emphasizing mostly on the coupled expression of PhANGs and PhAPGs by providing a novel mechanism that allows the flexibility to uncouple PhANGs and

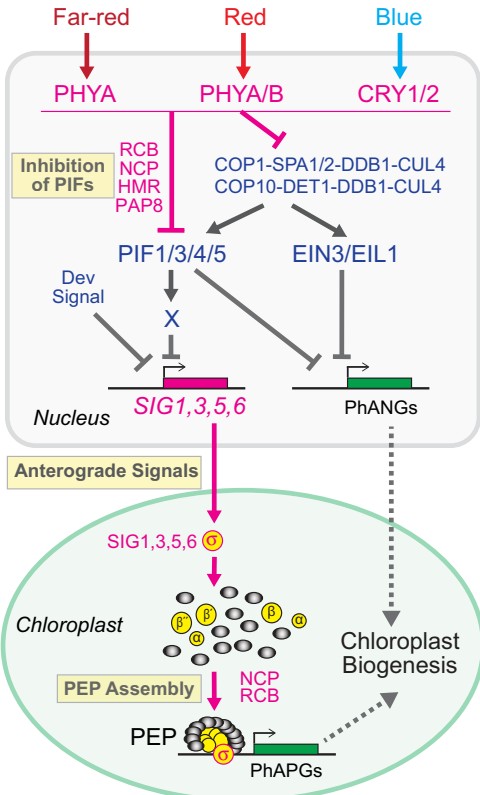

**Fig. 9 | Framework of anterograde signaling for the nuclear control of plastid transcription.** Monochromatic far-red, red, and blue light – which are perceived by PHYA, PHYA/PHYB (PHYA/B), and CRY1/CRY2 (CRY1/2), respectively – can individually initiate anterograde signaling by blocking the accumulation and activity of PIF1, PIF3, PIF4, and PIF5. PHYs and CRYs inhibit PIFs either directly via physical interaction or indirectly by attenuating the activity of COP1-SPA- and COP10-DET1-containing CUL4-based E3 ubiquitin ligases, which stabilize PIFs and also EIN3/EIL1. PIFs simultaneously repress both PhANGs and PhAPGs via parallel mechanisms. While PIFs work with EIN3/EIL1 to repress PhANGs in the nucleus, PIFs alone separately control the activation of the PEP in plastids by repressing the expression of four light-inducible SIGs, SIG1, SIG3, SIG5, and SIG6. PIFs likely repress SIGs indirectly via a yet unknown PIF-induced factor X. During early seedling establishment, anterograde signaling is suppressed by a developmental signal independently of PIFs. Several dual-targeted proteins, including RCB, NCP, HMR/PAP5/pTAC12 and PAP8/pTAC6, participate in the regulation of anterograde signaling in both the nucleus and plastids.

PhAPGs in response to developmental and environmental cues. The expression of PhANGs and PhAPGs has been widely accepted to be "coupled" primarily because of their synchronous expression along with the repression and activation of chloroplast biogenesis in the dark and light, respectively. We now know that the synchronization of PhANGs and PhAPGs by light is achieved by the regulation of the master nuclear regulators PIFs (Fig. 9). However, how PIFs coordinate the expression PhANGs and PhAPGs remained unclear. There are two possible mechanisms. PIFs could synchronize PhANGs and PhAPGs via a hierarchical mechanism, in which the activation of PhANGs itself serves as an anterograde signal to induce PhAPGs. Alternatively, PIFs could regulate PhANGs and PhAPGs via parallel mechanisms. Here, by examining precisely defined PhANGs in de-etiolated mutants, we showed that the genome-wide activation of PhANGs is insufficient to elicit PEP assembly and PhAPG activation, suggesting that PhANGs and PhAPGs are regulated by genetically separable, parallel mechanisms (Fig. 9). This conclusion is supported by the distinct regulation of PhANGs and PhAPGs by EIN3/EIL1 (Fig. 2) and a developmental signal (Fig. 3). Corroborating this idea, the ectopic expression of LHCBs, which has been shown to promote the formation of prothylakoid

membranes in plastids in darkness, was unable to trigger PEP assembly or PhAPG activation (Fig. 4).

Our model of a paralleled regulation of PhANGs and PhAPGs by PIFs provides the mechanistic basis to explain the observations of uncoupled expression of PhANGs and PhAPGs. The expression of PhANGs and PhAPGs is also coupled by the retrograde GUN (Genome UNcoupled) signaling[4]. The current model of GUN signaling posits that the PEP activity generates a signal that promotes the expression PhANGs[52]. As such, disrupting plastid transcription downregulates PhANGs. However, in the GUN signaling mutant gun1, the expression of PhANGs remains high even when plastid transcription is blocked[4]. Intriguingly, we showed that the same type of uncoupled expression of PhANGs and PhAPGs also occurred during early seedling development (Figs. 3 and 7). Our results not only provide a mechanistic explanation for the genome uncoupled phenomenon in gun mutants but also suggest that an uncoupled expression of PhANGs and PhAPGs represents a temporal state in plant development. Recently, the uncoupled expression of PhANGs and PhAPGs was shown to be required for generating reactive oxygen species to mediate salicylic acid signaling[59]. PhANGs and PhAPGs can be uncoupled by SIGMA FACTOR BINDING PROTEIN 1, which is dual-targeted to the nucleus and chloroplasts to induce PhANGs and repress PhAPGs[59]. The differential expression of PhANGs and PhAPGs results in an imbalance between photosystems II and I, thereby generating singlet oxygen species to trigger programmed cell death[59]. These observations further support the idea that the genetic architecture of the paralleled control of PhANGs and PhAPGs may play a profound functional role in cell signaling and adaptation to abiotic and biotic stress.

We devised a strategy that identified SIG1, SIG3, SIG5, and SIG6 as anterograde signals downstream of PIFs (Fig. 9). The light-dependent regulation of SIG1, SIG3, SIG5, and SIG6 as well as their critical functions in plastid transcription have been extensively reported[16,60–62]. The contribution of this work is to provide evidence distinguishing them as anterograde signals from the other PEP components. The first clue supporting this conclusion came from the correlation between the induction of these SIGs and PEP assembly and activation in de-etiolated mutants. The expression of SIG1, SIG3, SIG5, and SIG6 correlates with the pattern of PEP activation – i.e., they were activated in 4-d-old pifq but remained inactive in 4-d-old ein3/eil1 and 2-d-old pifq (Fig. 7a). These PIF-repressed SIGs were rapidly induced by light during the dark-to-light transition (Fig. 8c), further supporting their role as anterograde signals. Our conclusion is consistent with the recent findings that the circadian and light-intensity dependent expression of plastid photosynthesis genes is controlled by the nucleus via SIG5[63,64]. Our results showed that PIF3 represses PhAPGs through its activator activity (Fig. 6). Therefore, PIF3 and possibly also other PIFs likely repress SIGs indirectly by promoting the expression of a repressor protein X (Fig. 9).

Our results indicate that PIFs do not regulate PAPs at the transcript level (Supplementary Fig. 1)[43]. These results contradict the studies using cultured Arabidopsis cell lines, which showed that the expression of some PAPs was under the control of PIFs and the PEP complex became fully assembled in etioplasts[65,66]. These discrepancies are likely due to the different experimental model and/or assay conditions. For example, sucrose had to be used to sustain the growth of Arabidopsis cell culture, whereas, in our experiments, no supplemental sucrose was used because sucrose is known to influence light signaling and photosynthesis gene expression significantly[67]. It is important to note that we have focused on identifying PIF-dependent anterograde signals. Light also induces the expression of PIF-independent PEP components, including SIG2, SIG4, PAP2, PAP10, and PAP11 (Fig. 8c). The regulation of SIG2 by PHYs has been extensively reported[68]. SIG2 is involved in the regulation of PHY signaling, including PHY-dependent hypocotyl elongation and the expression of PhANGs[68,69]. Therefore, it is highly likely that the PIF-independent PEP components, such as SIG2,

also contribute to the light-dependent activation of the PEP via PIF-independent mechanisms.

The current data indicate that SIG1, SIG3, SIG5, and SIG6 are not the only anterograde signals downstream of PIFs. Similar to bacterial SIGs, plastid SIGs are most likely a dispensable subunit not required for the assembly of the PEP complex (Fig. 7b, c and Supplementary Fig. 2). Therefore, the light-dependent assembly of the PEP must be regulated by a yet unidentified anterograde signal. Two good candidates are REGULATOR OF CHLOROPLAST BIOGENESIS (RCB) and NUCLEAR CONTROL OF PEP ACTIVITY (NCP), which are dual-targeted proteins that participate in PHY-mediated PIF regulation in the nucleus and PEP assembly in plastids (Fig. 9)[43,44,70]. Also, we cannot exclude the possibility that some PAPs are involved in the regulation of PEP assembly by light. Although PAPs are not regulated by PIFs at the transcript level, they could still be regulated by light at the post-translational level. In fact, PHYB promotes the accumulation of HMR/PAP5/pTAC12 via direct interaction[71]. Moreover, because HMR/PAP5/pTAC12 and PAP8/pTAC6 are also dual-targeted to the nucleus and plastid and participate in both nuclear PHY signaling and PEP assembly (Fig. 9)[72–74], their subcellular partitioning could be another mechanism to modulate PEP assembly. Future investigations will focus on dissecting the anterograde signal for PEP assembly.

Our new findings substantially advanced the understanding of anterograde signaling in the control of plastid transcription by light. We demonstrated that, in addition to monochromatic red light, far-red and blue light – perceived by PHYA and CRYs, respectively – can individually initiate anterograde signaling to turn on the PEP and the expression of PhAPGs (Figs. 8 and 9). These results are consistent with the fact that CRYs regulate the stability and activity of PIFs in a similar manner as PHYs[12,53]. Arabidopsis seedlings grown in far-red light lack chlorophyll because POR, the enzyme catalyzing the conversion of protochlorophyllide to chlorophyll a, requires light with a shorter wavelength than far-red for its activity[8]. Therefore, the control of plastid transcription by anterograde signaling operates independently of chlorophyll biosynthesis and photosynthesis. This conclusion is further supported by the PEP activation in de-etiolated mutants in darkness (Fig. 1). It is also important to note that, although the general principle of anterograde signaling is expected to be the same under various light conditions, the extent of activation of the PIF-repressed SIGs was clearly different under distinct monochromatic light conditions (Fig. 8c), which may provide an explanation for the light quality-dependent regulation of plastid gene expression, such as the activation of a unique blue-light responsive promoter of psbD by SIG5[75].

In conclusion, our study further elucidates the light signaling framework that coordinates nuclear and plastid transcription during chloroplast biogenesis. We identified four SIGs as anterograde signals and surprisingly unveiled that anterograde signaling operates in parallel with the regulation of PhANGs. The latter provides a conceptual basis for the uncoupled regulation of PhANGs and PhAPGs in response to developmental and environmental stimuli. Our study also reveals gaps in anterograde signaling for future investigations, including a developmental mechanism that gates the anterograde signal during early seedling development and a yet-unidentified mechanism for controlling PEP assembly by light.

## Methods
### Plant materials, growth conditions, and hypocotyl measurements
Arabidopsis Colombia (Col-0) was used as the wild-type control for PEP complex assembly and gene expression under various light conditions. The mutant lines, including pifq[17], cop1-4[45], det1-1[46], spa234[26], spa134[26], spa124[26], spa123[26], ein3/eil1[47], LHCB1.1ox[18], LHCB2.1ox[18], pif1[17], pif3[17], pif4[17], pif5[17], pif345[17], pif145[17], pif135[17], pif134[27], PIF3/pifq (1-2 and 9-5)[11], PIF3mAD/pifq (2-1 and 4-5)[11], sig1-1[52], sig2-2[52], sig3-2[52], sig4-2[52], sig5-3[52],

sig6-1[52], phyA-211[55], and cry1/cry2[76], have been previously reported. Seeds were surface sterilized in 50% bleach with 0.01% Triton X-100 for 10 min and then washed four times with ddH$_2$O. Seeds were then plated on half-strength Murashige and Skoog (½ MS) medium containing Gamborg's vitamins (MSP0506, Caisson Laboratories), 0.5 mM MES (pH 5.7), and 0.8% (w/v) agar (A038, Caisson Laboratories). Seeds were stratified in the dark at 4 °C for five days before being placed in an LED chamber (Percival Scientific) under the indicated conditions for four days. For the dark-grown samples, seeds were exposed to far-red light for 3 h to trigger germination before placing them back into the dark for an additional 93 h. Light intensity was measured using an Apogee PS200 spectroradiometer (Apogee Instruments).

### RNA extraction and quantitative real-time PCR
Total RNA from seedlings of the indicated genotypes and growth conditions was isolated using a Quick-RNA MiniPrep kit with on-column DNase I treatment (Zymo Research). cDNA was synthesized from total RNA using a Superscript II First-Strand cDNA Synthesis kit (Thermo Fisher Scientific) according to the manufacturer's protocol. Oligo(dT) primers were used for the analysis of nuclear gene expression, and a mixture of plastidial-gene-specific primers was used for the analysis of plastidial genes. qRT-PCR was performed with Bio-Rad iQ SYBR Green Supermix on a Roche LightCycler 96 system. The mRNA level of each gene was normalized to that of PP2A. Primers for qRT-PCR are listed in Supplementary Tables 3 and 4.

### Blue-native gel electrophoresis for analyzing PEP assembly
PEP assembly was analyzed via blue-native polyacrylamide gel electrophoresis (BN-PAGE) using a NativePAGE Sample Prep kit and a NativePAGE Novex Bis-Tris Gel system (Thermo Fisher Scientific) with immunoblot[77]. One hundred milligrams of seedlings grown under the indicated conditions was ground in liquid nitrogen and resuspended in 3 volumes of BN lysis buffer (100 mM Tris-Cl, pH 7.2; 10 mM MgCl$_2$; 25% glycerol; 1% Triton X-100; 10 mM NaF; 5 mM β-mercaptoethanol; 1× EDTA-free protease inhibitor cocktail). Protein extracts were divided into two tubes, one for BN-PAGE and the other for SDS-PAGE. For BN-PAGE, protein extracts were mixed with BN sample buffer (1× NativePAGE sample buffer, 50 mM 6-aminocaproic acid, 1% n-dodecyl β-D-maltoside (DDM), and Benzonase nuclease) and incubated for 60 min at room temperature to degrade DNA/RNA and solubilize the PEP complex. Samples were mixed with 0.25% NativePAGE Coomassie blue G-250 sample additive and centrifuged at 17,500 × g for 10 min at 4 °C. Proteins from the supernatant were separated on a 4–16% NativePAGE Bis-Tris protein gel using a NativePAGE Running Buffer kit (Thermo Fisher Scientific) according to the manufacturer's protocol and with the following modifications. NativeMark Unstained Protein Standard (Thermo Fisher Scientific) was used to determine protein size in BN-PAGE. Briefly, electrophoresis was performed at 30–40 V for 3 h at 4 °C until the blue dye migrated through one third of the gel. The Dark Blue Cathode Buffer was replaced with Light Blue Cathode Buffer, and electrophoresis was continued at 20–25 V overnight at 4 °C. After electrophoresis was complete, the separated proteins were transferred onto a polyvinylidene difluoride (PVDF) membrane using 1× NuPAGE Transfer Buffer (Thermo Fisher Scientific, Waltham, MA) at a constant 70 V for 7 h at 4 °C. After transfer, the membrane was fixed with fixation buffer (25% methanol, 10% acetic acid) for 15 min and washed with water. The membrane was incubated with methanol for 5 min to destain the dye, and then it was washed with water and immunoblotting continued. The membrane was blocked with 2% non-fat milk in 1× TBS (20 mM Tris-Cl pH 7.6, 150 mM NaCl), probed with the indicated monoclonal mouse anti-rpoB antibodies (PHY1700, PhytoAB Inc.), washed with 1× TBS containing 0.05% Tween-20, and then incubated with anti-mouse secondary antibodies conjugated with horseradish peroxidase (1706516, Bio-Rad). Primary and secondary antibodies were used at 1:1000 and 1:5000 dilutions, respectively. The signals were

detected using SuperSignal West Dura Extended Duration Chemiluminescent Substrate (Thermo Fisher Scientific).

## Protein extraction and immunoblot analysis

Total protein was extracted from *Arabidopsis* seedlings grown under the indicated conditions. Plant tissues were ground in liquid nitrogen and resuspended in extraction buffer (100 mM Tris-HCl pH 7.5, 100 mM NaCl, 1% SDS, 5 mM EDTA pH 8.0, 20 mM DTT, 40 µM MG132, 40 µM MG115, and 1× EDTA-free protease inhibitor cocktail), boiled for 10 min and then centrifuged at $20,000 \times g$ for 10 min at room temperature. Protein extracts were separated via SDS-PAGE, transferred to nitrocellulose membranes, probed with the indicated primary antibodies, and then incubated with HRP-conjugated secondary antibodies. Mouse monoclonal anti-actin (A0480, Sigma-Aldrich) was used at a 1:4000 dilution, and anti-RPN6 (BML-PW8370-0100, Enzo Life Sciences) was used at a 1:1000 dilution. Goat anti-mouse (1706516, Bio-Rad) and anti-rabbit (1706515, Bio-Rad) secondary antibodies were used at a 1:5000 dilution. Signals were detected via SuperSignal West Dura Extended Duration Chemiluminescent Substrate (Thermo Fisher Scientific).

## RNA-seq data analysis

RNA-seq data analysis was performed with the pRNASeqTools pipeline (https://github.com/grubbybio/pRNASeqTools). Briefly, the raw reads were mapped to the Araport11[78] genome using STAR version 2.7.7a[79], and the number of reads mapped uniquely to each annotated gene was counted using FeatureCounts version 2.0.1[80]. Transcript levels were measured in fragments per kilobase per million total mapped fragments (FPKM). Differentially expressed genes were identified using DEseq2 version 3.14[81] with a fold change of two and $P < 0.01$ as the parameters. The relative transcript levels of PhANGs were normalized to that of *PP2A*.

## Reporting summary

Further information on research design is available in the Nature Portfolio Reporting Summary linked to this article.

## Data availability

*Arabidopsis* mutants used in the current study are available from the corresponding author. The published RNA-seq data used in this study are listed in Supplementary Table 2. The source data underlying Figs. 1a–b, 2c–d, 3a–b, 3e, 4a–b, 5a–c, 6a–b, 7a–c, 8a–c, Supplementary Fig. 1, and Supplementary Fig. 2a–b are provided as a Source Data file. Source data are provided with this paper.

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

## Acknowledgements
We are grateful to Joanne Chory (Salk Institute) for providing the *det1-1* mutant, Xing Wang Deng (Peking University) for providing the *cop1-4* mutant, Peter Quail for sharing the *pifq* mutant, Hong Qiao (University of Texas) for providing the *ein3/eil1* mutant, Jesse Woodson (University of Arizona) for providing the *sig2-2* and *sig6-1* mutants, Beronda Montgomery (Michigan State University) for providing the *sig1-1, sig3-2, sig4-2,* and *sig5-3* mutants, and Chentao Lin (University of California) for providing the *cry1/cry2* mutant. We thank Thomas Girke, Jordan Hayes, and the UCR High-Performance Computing Center for providing assistance with the bioinformatics tools used in the study. We thank Elise Pasoreck for valuable suggestions and comments on the manuscript. This work was supported by National Institute of General Medical Sciences grant R01GM132765 to M.C. and National Science Foundation grant IOS-2034015 to X.C. and M.C.

## Author contributions
Y.H., S.H., C.Y.Y., X.C. and M.C. conceived of the original research plan; M.C. supervised the experiments; Y.H., S.H., C.Y.Y., L.H. and M.C. performed the experiments; H.S. and S.Z. provided the *LHCB1.1ox* and *LHCB2.1ox* lines; U.H. provided the spa mutants; Y.H., S.H., C.Y.Y., L.H., C.Y., B.H.L., H.S., S.Z., U.H., X.C., and M.C. analyzed the data; Y.H. and M.C. wrote the article with contributions from all authors.

## Competing interests
The authors declare no competing interests.
