## [Peer Review File · Nature Communications]

Anterograde signaling controls plastid transcription via sigma factors separately from nuclear photosynthesis genesReviewer #1 (Remarks to the Author):

In the manuscript "Light synchronizes chloroplast transcription with the nuclear photosynthesis program in Arabidopsis via sigma factors as anterograde signals", authors found parallel activation of PhANGs and PhAPGs by studying de-etiolated mutants including *pifq*, *cop1-4*, *det1-1*, four *spa* triple mutants (*spa234*, *spa134*, *spa124*, and *spa123*), and *ein3/eil1*. These mutants constitutively upregulate the expression of nuclear photosynthesis genes in dark, however, are not sufficient to activate the PEP and PhAPGs expression. The authors discovered that the nuclear-encoded sigma factors correlate with the PEP activation and PhAPGs expression. PHYs and CRYs synchronize chloroplast transcription with the nuclear photosynthesis program through sigma factors that act as nucleus-to-plastid signals to initiate chloroplast biogenesis.

How nucleus regulates organellar gene expression in response to developmental and environmental cues is unclear, but is very important. The nucleus-to-organelle signaling, anterograde signaling, can be independent of the activation of nuclear photosynthesis genes and provides a research basis for the independent regulation of PhANGs and PhAPGs in response to developmental and environmental adaptation. Thus the subject is relevant and deserves to be published in an important journal such as Nature Communications upon clarification of the following points.

1. Line 242 and line 280: "119 of 146 PhANGs were upregulated in 4-d-old dark-grown *pifq*", while 87 of the 149 PhANGs were upregulated in 2-d-old dark-grown *pifq*. How did transcription levels of sigma factors change under these two conditions by analyzing published RNA-seq datasets?
2. Authors divided the PhANGs into three subcategories, including LHCs, CHLs and nPEACs. However, the transcripts level analysis in Fig 2-c and Fig 5-d only includes LHCs and nPEACs. What about CHLs?
3. Four sigma factors, including SIG1, SIG3, SIG5, and SIG6, were induced in 4-d-old *pifq*, while SIG2 and SIG4 are unchanged. However, the authors only used *sig2-2* and *sig6-1* to confirm that the absence of sigma factors affects PEP activation, which does not in line with the transcriptional level. Additional data of other sigma factor mutants will provide more evidence and tell more about it.

Reviewer #2 (Remarks to the Author):

Several of the results are rather unlikely:

The involvement of sigma factors in anterograde signaling and in assembly of the PEP is improbable. Indeed, these results are not only in opposition to the function of sigma factors in bacteria (Feklistov et al. 2014, *Ann. Rev. Microbiol* 868: 357-376), but they also in conflict with data about chloroplast development published by Joanne Chory and co-workers, i.e. the paper of Woodson et al. (2014), *Plant Journal* 73, 1-13. In the latter paper the involvement of sigma factors in retrograde signaling has been demonstrated.

The authors write that by their proposed mechanism involving sigma factors as PEP assembly factors, plastid gene expression and expression of nuclear encoded genes for photosynthesis (PHANGS) are activated in parallel. In contrast, former studies have shown that the synthesis of the plastid encoded reactions center of photosynthesis (encoded by plastid genes) precede the formation of light harvesting complexes which are nucleus encoded. This is in particular obvious under light limiting conditions (intermittent light) as described already by the group of Akoyunoglou in the eighties.

In this manuscript, RpoB has not been detected in samples prepared from proplastids and also etioplasts. This is rather unlikely. In contrast, in a recent paper on PEP formation (not cited by the authors) it is obvious that the PEP core components are already present in etioplasts from plants not exposed to light (Ji et al. 2021, *Physiol. Plant.* 171:435-446).

Reviewer #3 (Remarks to the Author):

Light is an essential signal involved in the activation of the of PhANGs in the nucleus and the PhAPGs in the plastids to promote chloroplast development and functions. The light photoreceptors are known to set up the anterograde communication channels between the nucleus and the plastids to coordinate responses and promote environmental responsiveness, however the exact molecular mechanisms remain unclear.

This manuscript focuses on dissecting the nature of the anterograde signals controlled by light that promote the transcription of plastid encoded genes by the plastid encoded polymerase (PEP). It also addresses whether the activation by light of PhANGs and PhAPGs in hierarchical or rather involves a parallel mechanism that permits uncoupling. The experimental results support the view that the global light-activation PhANGs controlled by the de-repression caused by the degradation of the phytochrome interacting factors, is not sufficient to promote the assembly of the PEP in the plastids and promote active PhAPG transcription during deetiolation. Results indicate a new functional role for the sigma factors as the light-modulated anterograde signal that promotes PEP assembly in the plastids with the up-regulation of the PhAPGs.

Interestingly, this anterograde signaling can work in parallel to the nuclear activation of PhAPGs, allowing the uncoupling of responsiveness, what may be necessary under natural environments or as part of the developmental responses during deetiolation.

The dissection of the unknown molecular mechanisms behind the photoreceptors anterograde communication with the plastids to promote the transcription of the chloroplast genome is of high interest to address chloroplast functions and understand interorganellar communication. As such this research makes an important contribution to our understanding of the coordination capacity of the photoreceptors of interorganellar communication and function. The topic is of high interest to the plant science community, but also the relevance extends to deepen our understanding of interorganellar coordination.

While most of the conclusions are supported by the data presented, some comments on specific aspects follow:

1. The Title "synchronization of chloroplast transcription with the nuclear photosynthesis program via Sigma factors", I find it a bit miss leading, as it could be interpreted as sigma factors being involved in the activation of PhANGs and PhAPGs, but this is not the mechanism proposed. Sigma factors are rather defined as the anterograde signal in charge of activating the assembly of the PEP for PhAPGs transcription and light activation of PhANGs by photoreceptors is a parallel mechanism.

2. The authors propose that the sigma factors are controlling the PEP complex assembly to boost transcription of PhAPGs, but is it complex assembly or complex abundance what they are controlling? What is the evidence that the assembly is light regulated? (as the PEP is active in darkness, even it at low rates).

3. Is it possible that the sigma factors in deetiolation are involved in changing the balance between the NEP activity and the PEP activity, as part of the anterograde signaling proposed?

4. Does the contribution of the different sigma factors to PEP assembly/abundance, change with light quality? Is the SIG2 and SIG6 role in PEP modulation red-light specific? Or are we looking at a general mechanism set up by the different photoreceptors for anterograde signaling?

5. Beyond SIG2 and SIG6 in Red, are the other sigma factors involved in light quality perception working with phyA and/or CRYs, as these receptors are proposed to have a similar effect on the abundance of PEP?.

6. As the abundance of PEP complex is also controlled by the CRYs and phyA, is the independent effect of activation of phAPGs and phANGs also happening in blue and FR? Or is this uncoupling capacity only regulated by Red-phyB?. In other words, is it a phy specific mechanism or the authors are describing a general mechanism by used by the blue and red light photoreceptors to set up parallel pathways of nuclear and plastidic

gene expression activation?

7. What are the characteristics that factor X would have to fill, towards looking for potential candidates?

8. The sigma factors have been proposed by some authors as proteins that affect the *phANG* expression. While the mechanism is unclear, which percentage of the *phANGs* defined in this paper are affected in *sig2* and or *sig6*?. Since the Sigma Factors have been proposed by some authors as dual localized proteins, have the authors verified that under the deetiolation conditions used the factors are chloroplastic and no nuclear signal is detected?

9. Can the authors include the light conditions used for each experiment?. Not all the experiments include the details on light intensity and light quality. Most experiments involve Red responses, but if anything is done in a different condition, it would be appropriate to know it.

Reviewer #4 (Remarks to the Author):

This manuscript by Hwang et al. addresses the identity of the nucleus-to-chloroplast anterograde signaling leading to the synchronization of nuclear and PEP-mediated chloroplast transcription during light-induced chloroplast biogenesis. Sigma factors are nuclear-encoded PEP subunits that are known for quite a long time to be regulated by light through different photoreceptors, with direct involvement of PIFs. Here, authors explore a possible role of sigma factors as anterograde signals to initiate PEP-mediated chloroplast transcription during seedling deetiolation. The paper is well written, builds on previous solid work done in the laboratory, and data showing PEP assembly and gene expression in different *cop*-like mutant backgrounds is of quality, and overall leads authors to propose that sigma factors act as anterograde signals.

-I agree with the authors that it is challenging to identify nucleus-to-plastid signals because it is hard to distinguish between true regulators and those that are essential to chloroplast biogenesis. Because sigma factors are essential components of PEP, I feel this issue is still an unresolved concern.

-Although the authors present an array of data suggesting that sigma factors act as anterograde signals, little genetic evidence is presented in support of the proposed model. For example, if sigma factors are acting as anterograde signals as suggested, then we would expect that their expression in *Col-0* (and in *ein3ein1*) would induce PEP assembly in the dark. Moreover, we would expect that assembly of PEP does not take place in *pifqsig* mutants in the dark.

- Previously, the authors have shown that during the dark-to-R-light transition, the PEP complex appeared within 1 h after light exposure and increased to a steady-state level in 48 h (Yoo et al., 2019). I think it is critical to incorporate this temporal aspect into the current manuscript to support the proposed model, which now relies mostly on expression data in deetiolated mutants in the dark and levels of PEP complex in the dark or after long exposure to light. This would also help rule out possible secondary effects. How is the induction of SIG factors and PAPs during early deetiolation time points? What happens with PEP assembly in *sig* mutants at early time points of light exposure? It would also help to have PAP mutants added as controls.

- Authors discussed SIG2 expression is not significantly regulated in PIFs (fig 7a) (please correct line 486 in discussion) yet the *sig2* mutant is affected in 4d-old light-grown seedlings (fig 7b). Fig 7b shows PEP levels after 4 days of exposure to red light, a condition that it is not included in gene expression experiments. To be able to correlate PEP activation with the pattern of expression, can authors include expression of sigma factors and PAP under 2d/4d of R light compared to dark? Are SIG2 and SIG6 light-induced under these conditions?

-The Sigma factor most clearly affected in Fig 7a is SIG1, which makes it an obvious candidate for anterograde signal based on the authors' criteria. Have you tested PEP assembly in sig1 mutants?

-I suggest to include a section in the introduction with a comprehensive literature review on what it is currently known about the regulation of sigma factors by light and their proposed function in plants. I missed several published works that are necessary for completion and would help to provide context.

Response to Reviewers

To all reviewers

We thank you all for your positive comments and valuable suggestions. We wanted to particularly point out that we have revised our conclusion regarding the role of sigma factors (SIGs) in the assembly of the plastid-encoded RNA polymerase (PEP). During the revision, we realized that our original quantification method for the abundance of the PEP complex was not particularly suitable for assessing PEP assembly. Therefore, we modified our quantification method for evaluating PEP assembly. Based on the revised quantification method and the new experimental data, we concluded that plastid SIGs are not required for PEP assembly. The following is a detailed explanation.

An improved quantification method for the relative PEP level led to the conclusion that plastid SIGs are NOT required for the assembly of the PEP complex. All reviewers commented on the role of SIGs in PEP assembly. These comments pushed us to characterize the assembly of the PEP complex in all six *sig* mutants. During these new experiments, we realized that our original quantification method for the abundance PEP complex does not specifically reflect PEP assembly per se. Thus, we modified our method to quantify the relative PEP complex (in all figures). Our new conclusion is based on the new method.

We have been using immunoblots for the core PEP subunit rpoB to quantify the formation of the PEP complex. In our assays, we split the plant materials equally into two fractions and isolated proteins using either native or denatured conditions. The native samples were resolved using blue-native PAGE to detect rpoB in the 1000-kDa PEP complex and the denatured samples were resolved using SDS-PAGE to detect the total rpoB. We had previously (in the original manuscript) quantified the PEP complex by normalizing the intensity of the rpoB band in the blue-native PAGE (the PEP complex) using the loading control actin or RPN6. Although this method quantifies the relative abundance of the PEP complex between samples, it does not accurately assess PEP assembly, because the relative abundance of the PEP complex in the blue-native PAGE is also determined by the total rpoB. Therefore, less rpoB in the blue-native gel does not necessarily mean compromised PEP assembly, as it could also be attributed to a reduction in total rpoB. The original method might still work if we were only to assess the formation of the PEP complex qualitatively. But it became a problem when we wanted to assess PEP assembly quantitatively.

To circumvent the problem, we modified our quantification method by normalizing the rpoB band in the PEP complex (blue-native gel) using total rpoB (the band in SDS-PAGE). We named this value the relative PEP level. We reanalyzed PEP assembly in all Figures using this method. Also, we performed new experiments to assess PEP assembly during seedling development of Col-0 and in all 6 *sig* mutants (Fig. 8b,c and Supplementary Fig. 2). The new

results indicate that SIGs are not required for PEP assembly. However, because the SIGs are necessary subunits of the PEP holoenzyme for promoter targeting and transcription initiation, our results thus suggest that light controls the activity of the PEP via the PIF-repressed SIGs as anterograde signals.

Reviewer #1

Reviewer #1 (Remarks to the Author):

In the manuscript “Light synchronizes chloroplast transcription with the nuclear photosynthesis program in Arabidopsis via sigma factors as anterograde signals”, authors found parallel activation of PhANGs and PhAPGs by studying de-etiolated mutants including pifq, cop1-4, det1-1, four spa triple mutants (spa234, spa134, spa124, and spa123), and ein3/eil1. These mutants constitutively upregulate the expression of nuclear photosynthesis genes in dark, however, are not sufficient to activate the PEP and PhAPGs expression. The authors discovered that the nuclear-encoded sigma factors correlate with the PEP activation and PhAPGs expression. PHYs and CRYs synchronize chloroplast transcription with the nuclear photosynthesis program through sigma factors that act as nucleus-to-plastid signals to initiate chloroplast biogenesis.

How nucleus regulates organellar gene expression in response to developmental and environmental cues is unclear, but is very important. The nucleus-to-organelle signaling, anterograde signaling, can be independent of the activation of nuclear photosynthesis genes and provides a research basis for the independent regulation of PhANGs and PhAPGs in response to developmental and environmental adaptation. Thus the subject is relevant and deserves to be published in an important journal such as Nature Communications upon clarification of the flowing points.

1. Line 242 and line 280: “119 of 146 PhANGs were upregulated in 4-d-old dark-grown pifq”, while 87 of the 149 PhANGs were upregulated in 2-d-old dark-grown pifq. How did transcription levels of sigma factors change under these two conditions by analyzing published RNA-seq datasets?

Response: We thank the reviewer for the comments. We used the published datasets to analyze the genome-wide expression of PhANGs. One potential caveat of the published datasets was that the experiments were performed using slightly different seedling growth conditions. For example, some of the studies used seedlings grown on media containing sucrose, which could influence the results (described in Supplementary Table 2). So, we verified the RNA-seq data using qRT-PCR (Figs. 2 and 3). For the same reason, when we evaluated the expression of PAPs and SIGs, we performed qRT-PCR experiments instead of simply using the RNA-seq data – as, in our opinion, our data are more reliable. To answer the reviewer’s question, we went back and analyzed the expression of the six SIG genes in the RNA-seq data, the results are shown below. In the RNA-seq data, SIG2 and SIG6 were induced only in 4-d-old pifq, and SIG1, SIG3, and SIG5 were upregulated in both the 2-d- and 4-d-old pifq. Compared these results with our qRT-PCR data, only SIG6 qualified to be an anterograde signal in both. However, even with the new information, we still think we should draw our conclusion based on our qRT-PCR results.

2. Authors divided the PhANGs into three subcategories, including LHCs, CHLs and nPEACs. However, the transcripts level analysis in Fig 2-c and Fig 5-d only includes LHCs and nPEACs. What about CHLs?

Response: This is because we did not include genes involved in chlorophyll biosynthesis in our RNA-seq analysis in the early stage of the work. But, later on, to make PhANGs more comprehensive, we added the CHLs to the analysis. So, the verification qRT-PCR experiments performed at the early stage of the project used genes from the nPEACs. But, we do not expect that the choice of representative genes would affect our conclusion.

3. Four sigma factors, including SIG1, SIG3, SIG5, and SIG6, were induced in 4-d-old pifq, while SIG2 and SIG4 are unchanged. However, the authors only used sig2-2 and sig6-1 to confirm that the absence of sigma factors affects PEP activation, which does not in line with the transcriptional level. Additional data of other sigma factor mutants will provide more evidence and tell more about it.

Response: Please see our response to all reviewers at the beginning of this document. Briefly, we have revised our quantification method for PEP assembly and performed new experiments to evaluate PEP assembly in all 6 *sig* mutants (Fig. 7b.c and Supplementary Fig. 2). The new results indicate that SIGs are not required for PEP assembly. However, because SIGs are known as necessary subunits of the PEP for promoter targeting and transcription initiation, our results suggest that the PIF-repressed SIGs could act as anterograde signals to control the activity of the PEP.

Reviewer #2 (Remarks to the Author):

Several of the results are rather unlikely:

The involvement of sigma factors in anterograde signaling and in assembly of the PEP is improbable. Indeed, these results are not only in opposition to the function of sigma factors in bacteria (Feklistov et al. 2014, Ann. Rev. Microbiol 868: 357-376), but they also in conflict with data about chloroplast development published by Joanne Chory and co-workers, i.e. the paper of Woodson et al. (2014), Plant Journal 73, 1-13. In the latter paper the involvement of sigma factors in retrograde signaling has been demonstrated.

Response:

We thank the Reviewer for the comments. There are two questions here. The first question is regarding the role of plastid SIGs in PEP assembly. Please see our response to all reviewers at the beginning of the document. We revised the quantification method and included new data (Fig. 7b.c and Supplementary Fig. 2). Our new results agree with the reviewer's comment that plastid SIGs are not required for PEP assembly.

The second question is regarding our conclusion in relation to the retrograde GUN signaling. The data presented in this study do not contradict the theory of the retrograde GUN signaling pathway. On the contrary, our model of a parallel regulation of PhANGs and PhAPGs provides the mechanistic basis to explain the genome-uncoupled phenomenon in the *gun* mutants. We have largely revised the Discussion to highlight the significance of the discovery that PhANGs and PhAPGs are controlled by genetically separable mechanisms. Our conclusion that the four PIF-repressed SIGs act as anterograde signals also does not conflict with the data presented by Woodson et al., as we focused on their roles in anterograde signaling and the control by PHYs in the nucleus. We have to point out that the Woodson et al. study showed that perturbation of the PEP activity in *sig2* and *sig6* mutants could trigger the retrograde signaling response, which does not necessarily mean SIG2 and SIG6 are directly involved in retrograde signaling.

The authors write that by their proposed mechanism involving sigma factors as PEP assembly factors, plastid gene expression and expression of nuclear encoded genes for photosynthesis (PHANGS) are activated in parallel. In contrast, former studies have shown that the synthesis of the plastid encoded

reactions center of photosynthesis (encoded by plastid genes) precede the formation of light harvesting complexes which are nucleus encoded. This is in particular obvious under light limiting conditions (intermittent light) as described already by the group of Akoyunoglou in the eighties.

Response: We think the reviewer referred to the study by G. Tzinis et al. (1987) published in *Photosynth. Res.* This study tested whether the amount of the LHC complexes incorporated into the thylakoid during de-etiolation depended on the chlorophyll content. The results suggested that “it is not the chlorophyll content per se which regulates the stabilization of LHC in development thylakoids and consequently the size of the PS unit, but rather the rate by which it is accumulated, relative to that of the other thylakoid components.” The authors measured the amount of PSII and PSI but not gene expression (accumulation of mRNAs). This is quite important because the activation of PhANGs and PhAPGs described in our study is independent of photosynthesis, because they occur in the de-etiolated mutants in darkness (Figs. 1 and 2) and in Col-0 in far-red light (Fig. 8). The anterograde signaling mechanisms can clearly occur before the functional assembly of photosynthetic apparatus. Therefore, we do not think the studies by the Akoyunoglou group in the eighties addressed the relationship between nuclear and plastid gene expression.

In this manuscript, RpoB has not been detected in samples prepared from proplastids and also etioplasts. This is rather unlikely. In contrast, in a recent paper on PEP formation (not cited by the authors) it is obvious that the PEP core components are already present in etioplasts from plants not exposed to light (Ji et al. 2021, Physiol. Plant. 171:435-446).

Response: We thank the Reviewer for this comment. We partially addressed this discrepancy in our previous submission, as we cited an earlier publication from the same authors. We have included the new reference in the revised manuscript. Please see our explanation in the Discussion below:

“Our results indicate that PIFs do not regulate *PAPs* at the transcript level (Supplementary Fig. 1)⁴³. These results contradict the studies using cultured *Arabidopsis* cell lines, which showed that the expression of some *PAPs* was under the control of PIFs and the PEP complex became fully assembled in etioplasts^{62,63}. These discrepancies are likely due to the different experimental model and/or assay conditions. For example, sucrose had to be used to sustain the growth of *Arabidopsis* cell culture, whereas, in our experiments, no supplemental sucrose was used because sucrose is known to influence light signaling and photosynthesis gene expression significantly⁶⁴.”

Reviewer #3 (Remarks to the Author):

Light is an essential signal involved in the activation of the of PhANGs in the nucleus and the PhAPGs in the plastids to promote chloroplast development and functions. The light photoreceptors are known to set up the anterograde communication channels between the nucleus and the plastids to coordinate

responses and promote environmental responsiveness, however the exact molecular mechanisms remain unclear.

This manuscript focuses on dissecting the nature of the anterograde signals controlled by light that promote the transcription of plastid encoded genes by the plastid encoded polymerase (PEP). It also addresses whether the activation by light of PhANGs and PhAPGs in hierarchical or rather involves a parallel mechanism that permits uncoupling.

The experimental results support the view that the global light-activation PhANGs controlled by the de-repression caused by the degradation of the phytochrome interacting factors, is not sufficient to promote the assembly of the PEP in the plastids and promote active PhAPG transcription during deetiolation. Results indicate a new functional role for the sigma factors as the light-modulated anterograde signal that promotes PEP assembly in the plastids with the up-regulation of the PhAPGs. Interestingly, this anterograde signaling can work in parallel to the nuclear activation of PhAPGs, allowing the uncoupling of responsiveness, what may be necessary under natural environments or as part of the developmental responses during deetiolation.

The dissection of the unknown molecular mechanisms behind the photoreceptors anterograde communication with the plastids to promote the transcription of the chloroplast genome is of high interest to address chloroplast functions and understand interorganellar communication. As such this research makes an important contribution to our understanding of the coordination capacity of the photoreceptors of interorganellar communication and function. The topic is of high interest to the plant science community, but also the relevance extends to deepen our understanding of interorganellar coordination.

While most of the conclusions are supported by the data presented, some comments on specific aspects follow:

1. The Title “synchronization of chloroplast transcription with the nuclear photosynthesis program via Sigma factors”, I find it a bit miss leading, as it could be interpreted as sigma factors being involved in the activation of PhANGs and PhAPGs, but this is not the mechanism proposed. Sigma factors are rather defined as the anterograde signal in charge of activating the assembly of the PEP for PhAPGs transcription and light activation of PhANGs by photoreceptors is a parallel mechanism.

Response: We thank the reviewer for the comment. This is a great point. We have modified the title to “Anterograde signaling controls plastid transcription via sigma factors in parallel with the regulation of nuclear photosynthesis genes”.

2. The authors propose that the sigma factors are controlling the PEP complex assembly to boost transcription of PhAPGs, but is it complex assembly or complex abundance what they are controlling? What is the evidence that the assembly is light regulated? (as the PEP is active in darkness, even it at low rates).

Response: We thank the reviewer for the comments. Please see our response to all reviewers at the beginning of the document. We now quantify PEP assembly by using the relative amount of rpoB in the PEP complex (blue-native PAGE) divided by the total rpoB (SDS-PAGE) – we call this the relative PEP level. As shown in Fig. 1a, although rpoB was present in dark-grown Col-0 (SDS-PAGE), the rpoB-containing PEP complex was not detectable, indicating that PEP

assembly was attenuated. This conclusion has been drawn in our previous publications [Yoo et al. Nat Commun (2019) 10:2630], but we believe our modified quantification is a better way to evaluate PEP assembly.

3. Is it possible that the sigma factors in deetiolation are involved in changing the balance between the NEP activity and the PEP activity, as part of the anterograde signaling proposed?

Response: We thank the reviewer for the comment. As reported previously, PEP-deficient mutants show elevated expression of NEP-regulated genes [Pfalz et al. Plant Cell (2006) 18:176-197]. One reason for this phenomenon is that the PEP-induced glutamyl-tRNA can directly bind and inhibit the activity of the NEP [Hanaoka et al. EMOB Rep (2005) 6:545-550]. So, light-triggered activation of the PEP does alter the balance between the PEP and NEP activities. But, this effect is a consequence as opposed to part of anterograde signaling.

4. Does the contribution of the different sigma factors to PEP assembly/abundance, change with light quality? Is the SIG2 and SIG6 role in PEP modulation red-light specific? Or are we looking at a general mechanism set up by the different photoreceptors for anterograde signaling? 5. Beyond SIG2 and SIG6 in Red, are the other sigma factors involved in light quality perception working with phyA and/or CRYs, as these receptors are proposed to have a similar effect on the abundance of PEP?.

Response: We thank the reviewer for the comments. This question is still valid, although the PIF-repressed SIGs only control PEP activity (but not assembly). The anterograde signals are triggered by the photo-inhibition of the stability and activity of PIFs. The paralleled regulation of PhANGs and PhAPGs described in our study happens downstream of PIFs. Because the current data indicate that PHY and CRY signaling mechanisms converge on the regulation of PIFs, it is expected that the general principle of anterograde signaling should be the same by the different photoreceptors. However, with that said, photoreceptor-specific regulation could also exist. In fact, it has been well documented, for example, blue light specifically induces the transcription of psbD from a unique blue-light responsive promoter via SIG5 [Tsunoyama et al. PNAS (2004) 101:3304-9]. One possible explanation is the extent of SIG induction is different under different photoreceptors, which is nicely shown in Fig. 8c. As a result, the amplitude of PhAPG expression could be different in different monochromatic light conditions (Fig. 8b). Since all SIGs are light-inducible (Fig. 8c), it is likely that all of them contribute to the expression of PhAPGs. We propose that besides the four PIF-repressed SIGs, other light-induced PEP components, including SIG2 and SIG4, may contribute to anterograde signaling via PIF-independent mechanisms.

6. As the abundance of PEP complex is also controlled by the CRYs and phyA, is the independent effect of activation of phAPGs and phANGs also happening in blue and FR? Or is this uncoupling capacity only regulated by Red-phyB?. In other words, is it a phy specific mechanism or the authors are describing a general mechanism by used by the blue and red light photoreceptors to set up parallel pathways of nuclear and plastidic gene expression activation?

Response: Please see our response above.

7. What are the characteristics that factor X would have to fill, towards looking for potential candidates?

Response: Factor X should be a PIF-activated transcriptional repressor directly binding to the promoters of *SIG1*, *SIG3*, *SIG5*, and *SIG6* to repress their expression in the dark. We cannot think of a good candidate at this point.

8. The sigma factors have been proposed by some authors as proteins that affect the phANG expression. While the mechanism is unclear, which percentage of the phANGs defined in this paper are affected in sig2 and or sig6?. Since the Sigma Factors have been proposed by some authors as dual localized proteins, have the authors verified that under the deetiolation conditions used the factors are chloroplastic and no nuclear signal is detected?

Response: This is a great idea. Studies from Prof. Beronda Montgonery revealed an interesting link between SIG2/6 and PHY signaling in the nucleus. It would certainly be very interesting to look into SIG2/6-dependent PhANGs and see how much they contribute to the branch of PhANGs. However, we think the focus of our current study is to dissect anterograde signals from among essential components of the PEP, it might be more appropriate to include the functional characterization of the SIGs in future investigations.

9. Can the authors include the light conditions used for each experiment?. Not all the experiments include the details on light intensity and light quality. Most experiments involve Red responses, but if anything is done in a different condition, it would appropriate to know it.

Response: We thank the reviewer for the comments. The general plant growth conditions are described in the Methods. We included specific growth conditions including seedling age and light conditions in the text and figure legends.

Reviewer #4 (Remarks to the Author):

This manuscript by Hwang et al. addresses the identity of the nucleus-to-chloroplast anterograde signaling leading to the synchronization of nuclear and PEP-mediated chloroplast transcription during light-induced chloroplast biogenesis. Sigma factors are nuclear-encoded PEP subunits that are known for quite a long time to be regulated by light through different photoreceptors, with direct involvement of PIFs. Here, authors explore a possible role of sigma factors as anterograde signals to initiate PEP-mediated

chloroplast transcription during seedling deetiolation. The paper is well written, builds on previous solid work done in the laboratory, and data showing PEP assembly and gene expression in different cop-like mutant backgrounds is of quality, and overall leads authors to propose that sigma factors act as anterograde signals.

-I agree with the authors that it is challenging to identify nucleus-to-plastid signals because it is hard to distinguish between true regulators and those that are essential to chloroplast biogenesis. Because sigma factors are essential components of PEP, I feel this issue is still an unresolved concern.

Response: We thank the reviewer for the comments. In our opinion, a major contribution of this work is the devised strategy to separate anterograde signals from among the 18 nuclear-encoded components of the PEP. We showed that our strategy – using transcriptomic analysis of the *pifq* mutant from two seedling stages in combination with the analysis of early light-responsive genes – could clearly separate light-inducible, PIF-repressed regulators from the rest of the essential components. Our strategy also clearly separated the regulation of PhANGs (which are essential for chloroplast biogenesis also) from anterograde signaling and the activation of PhAPGs.

*-Although the authors present an array of data suggesting that sigma factors act as anterograde signals, little genetic evidence is presented in support of the proposed model. For example, if sigma factors are acting as anterograde signals as suggested, then we would expect that their expression in Col-0 (and in *ein3ein1*) would induce PEP assembly in the dark. Moreover, we would expect that assembly of PEP does not take place in *pifqsig* mutants in the dark.*

Response: We thank the reviewer for the comments. As explained by our response to all reviewers, our new results show that SIGs are not required for PEP assembly. The current data suggest that SIGs are not the only anterograde signals. There must be an unidentified anterograde signal for promoting PEP assembly. Our unpublished results show that ectopic expression of SIG6 in the dark was not sufficient to induce PEP assembly and PhAPG induction, corroborating the idea that PEP activation requires other anterograde signals as well. We elaborated on this point in the Discussion section.

*- Previously, the authors have shown that during the dark-to-R-light transition, the PEP complex appeared within 1 h after light exposure and increased to a steady-state level in 48 h (Yoo et al., 2019). I think it is critical to incorporate this temporal aspect into the current manuscript to support the proposed model, which now relies mostly on expression data in deetiolated mutants in the dark and levels of PEP complex in the dark or after long exposure to light. This would also help rule out possible secondary effects. How is the induction of SIG factors and PAPs during early deetiolation time points? What happens with PEP assembly in *sig* mutants at early time points of light exposure? It would also help to have PAP mutants added as controls.*

Response: We thank the reviewer for the comments. As explained in our response to all reviewers, our new results indicate that SIGs are not required for PEP assembly. Nonetheless,

because SIGs are known as essential PEP components for promoter targeting and transcription initiation, our results suggest that the light-inducible PIF-repressed SIGs are anterograde signals to activate the PEP. With this conclusion, the reviewer's question was still valid as to whether the SIGs were induced rapidly by light. To address that question, we included RNA-seq data on the expression of the 18 nuclear-encoded components of the PEP during the dark-to-light transition (Fig. 8c). The results show that the four PIF-repressed SIGs were rapidly induced by light, supporting their role as anterograde signals.

- Authors discussed SIG2 expression is not significantly regulated in PIFs (fig 7a) (please correct line 486 in discussion) yet the sig2 mutant is affected in 4d-old light-grown seedlings (fig 7b). Fig 7b shows PEP levels after 4 days of exposure to red light, a condition that it is not included in gene expression experiments. To be able to correlate PEP activation with the pattern of expression, can authors include expression of sigma factors and PAP under 2d/4d of R light compared to dark? Are SIG2 and SIG6 light-induced under these conditions?

Response: We thank the reviewer for the comments. Based on the comments, we reanalyzed published RNA-seq data on 3-d-old dark-grown seedlings exposed to 1 h and 3 h of red, far-red, blue, and white light. Intriguingly, the 18 nuclear-encoded PEP components (12 PAPs and 6 SIGs) were grouped into two clusters: one induced by light and the other one repressed by light (Fig. 8c). All six SIGs were induced by light. Analyzing early light-responsive genes turned out to be a valuable means to confirm that the PIF-repressed anterograde signals are indeed light-inducible (Fig. 8c). Our results also show that some PIF-independent PEP components were induced by light, including SIG2, SIG4, PAP2, PAP10, and PAP11. We propose that these components could contribute to anterograde signaling via PIF-independent mechanisms.

-The Sigma factor most clearly affected in Fig 7a is SIG1, which makes it an obvious candidate for anterograde signal based on the authors' criteria. Have you tested PEP assembly in sig1 mutants?

Response: We thank the reviewer for the comments. Comments like this pushed us to look into the role of all SIGs in PEP assembly and to modify our quantification method for PEP assembly (Please see our response to all reviewers at the beginning of the document). Our new results (Fig. 7b,c and Supplementary Fig. 2) indicate that SIGs are not required for PEP assembly.

-I suggest to include a section in the introduction with a comprehensive literature review on what it is currently known about the regulation of sigma factors by light and their proposed function in plants. I missed several published works that are necessary for completion and would help to provide context.

Response: We added more discussion on sigma factors in both the Introduction and Discussion sections.

Reviewer #1 (Remarks to the Author):

The revised manuscript "Anterograde signaling controls plastid transcription via sigma factors in parallel with the regulation of nuclear photosynthesis genes" has addressed most of my concerns and reports important findings on the mechanism that light-dependent inhibition of PIFs activates plastid photosynthesis genes via sigma factors as anterograde signals. Authors performed more experiments, and more importantly, modified the quantification method for PEP assembly evaluation to reach more reasonable conclusions. The manuscript of current version could be recommendable to be published in Nature Communications.

Reviewer #3 (Remarks to the Author):

Understanding how the nucleus regulates organellar gene expression in response to environmental cues is a very relevant question to understand plant metabolism.

However, the mechanisms involved, including the nature of the anterograde signals and their links to photomorphogenesis remain unclear. In this manuscript the authors devise a way to distinguish anterograde regulators from essential chloroplast functioning components during deetiolation and provide with evidence that activation of PhANGs and PhAPGs can happen in parallel, as deetiolated mutants that have upregulation of photosynthetic gene expression in the dark do not show activation of the PEP and PhAPGs. Interestingly, the uncoupled expression of PhANGs and PhAPGs seems to be temporal, and may also be linked to environmental inputs.

The data provided also identifies PIFs as regulatory components that impose PEP inhibition and factors that repress light-induced sigma-factors that are essential for PEP activity and chloroplast transcription of PhAPGs. The investigation of the PIF-dependent anterograde pathways during deetiolation gives support to the role of sigma factors signaling components in the phy/cry photoreceptors cascades to the coordinate chloroplast transcription.

Although the new experiments indicate that plastid SIGs are not required for PEP assembly, this does not impact on the studies of SIGs as anterograde signals and on the light control of the PEP via PIF-repression. It also does not change the conclusion that PhANGs and PhAPGs can be uncoupled, and have a parallel modulation during deetiolation.

In the new version of the manuscript, the authors addressed the comments raised in the previous version, and the data presented is of quality and sustain the conclusions of sigma factors as anterograde components modulated by the light photoreceptors cascades. The data is of high quality and the topic is of relevance and will be of interest to the general audience, as well as open new avenues of investigation.

Some minor comments follow:

1. In the section "PIF-repressed sigma factors are anterograde signals" the sentence "the relative PEP level was only significantly reduced in 2-d old sig 6-1 but remained the same compared to Col-0"(Fig 7b) is confusing, do they mean "remained the same as Col-0 after 2d? (i.e. 3d, 4d). If so, maybe worth emphasizing here the time window associated.
2. In the section "PI-repressed sigma factors are rapidly induced by light", the manuscript refers to the PEP assembly being unaffected in single sig1-6 mutants", must be a typo and the authors meant sig 6-1 mutants.
3. In the same section, the authors analyzed the expression of 12 PAPs and 6 SIGs during light-dark transition and classified the PEP components in light up-regulated and light-down regulated. Based on the described functions of the PAPs, can the authors comment on the potential implications for the activity of the PEP holoenzyme of changes in PAPs modulated by light/dark?
4. There are a few typos to correct across the manuscript.

Reviewer #4 (Remarks to the Author):

The new experimental approach and the extensive revision provide solidity to the work and address my previous concerns

Response to Reviewers

Reviewer #1 (Remarks to the Author):

The revised manuscript “Anterograde signaling controls plastid transcription via sigma factors in parallel with the regulation of nuclear photosynthesis genes” has addressed most of my concerns and reports important findings on the mechanism that light-dependent inhibition of PIFs activates plastid photosynthesis genes via sigma factors as anterograde signals. Authors performed more experiments, and more importantly, modified the quantification method for PEP assembly evaluation to reach more reasonable conclusions. The manuscript of current version could be recommendable to be published in Nature Communications.

Response:

We thank the reviewer for the supporting comments.

Reviewer #3 (Remarks to the Author):

Understanding how the nucleus regulates organellar gene expression in response to environmental cues is a very relevant question to understand plant metabolism. However, the mechanisms involved, including the nature of the anterograde signals and their links to photomorphogenesis remain unclear. In this manuscript the authors devise a way to distinguish anterograde regulators from essential chloroplast functioning components during deetiolation and provide with evidence that activation of PhANGs and PhAPGs can happen in parallel, as deetiolated mutants that have upregulation of photosynthetic gene expression in the dark do not show activation of the PEP and PhAPGs. Interestingly, the uncoupled expression of PhANGs and PhAPGs seems to be temporal, and may also be linked to environmental inputs.

The data provided also identifies PIFs as regulatory components that impose PEP inhibition and factors that repress light-induced sigma-factors that are essential for PEP activity and chloroplast transcription of PhAPGs. The investigation of the PIF-dependent anterograde pathways during deetiolation gives support to the role of sigma factors signaling components in the phy/cry photoreceptors cascades to the coordinate chloroplast transcription.

Although the new experiments indicate that plastid SIGs are not required for PEP assembly, this does not impact on the studies of SIGs as anterograde signals and on the light control of the PEP via PIF-repression. It also does not change the conclusion that PhANGs and PhAPGs can be uncoupled, and have a parallel modulation during deetiolation.

In the new version of the manuscript, the authors addressed the comments raised in the previous version, and the data presented is of quality and sustain the conclusions of sigma factors as anterograde components modulated by the light photoreceptors cascades. The data is of high quality and the topic is of relevance and will be of interest to the general audience, as well as open new avenues of investigation.

Response:

We thank the reviewer for the supporting comments.

Some minor comments follow:

1. In the section “PIF-repressed sigma factors are anterograde signals” the sentence “the relative PEP level was only significantly reduced in 2-d old sig 6-1 but remained the same compared to Col-0”(Fig 7b) is confusing, do they mean “remained the same as Col-0 after 2d? (i.e. 3d, 4d). If so, maybe worth emphasizing here the time window associated.

Response:

We revised the sentence: “The relative PEP level was only significantly reduced in 2-d-old *sig6-1* but remained the same compared with Col-0 in 3-d and 4-d-old *sig6-1* (Fig. 7b).”

2. In the section “PI-repressed sigma factors are rapidly induced by light”, the manuscript refers to the PEP assembly being unaffected in single sig1-6 mutants”, must be a typo and the authors meant sig 6-1 mutants.

Response:

It meant the *sig1* to *sig6* single mutants. We modified the sentence to “ ... in the *sig* mutants...”

3. In the same section, the authors analyzed the expression of 12 PAPs and 6 SIGs during light-dark transition and classified the PEP components in light up-regulated and light-down regulated. Based on the described functions of the PAPs, can the authors comment on the potential implications for the activity of the PEP holoenzyme of changes in PAPs modulated by light/dark?

Response:

Although the 12 PAPs are essential components of the PEP, their precise functions remain less understood. Therefore, it would be difficult to suggest potential implications for the PEP function with the current data.

4. There are a few typos to correct across the manuscript.

Response:

We have edited the manuscript and corrected the typos.

Reviewer #4 (Remarks to the Author):

The new experimental approach and the extensive revision provide solidity to the work and address my previous concerns

Response:

We thank the reviewer for the supporting comment.